# Navigating the Design Space of Equivariant Diffusion-Based Generative Models for De Novo 3D Molecule Generation

**Tuan Le**[*,1,2], **Julian Cremer**[*,1,4],
**Frank Noé**[2,3], **Djork-Arné Clevert**[1], **Kristof Schütt**[1]

[1]Pfizer Research & Development, [2]Freie Universität Berlin
[3]Microsoft Research, [4]University Pompeu Fabra

## Abstract

Deep generative diffusion models are a promising avenue for 3D de novo molecular design in materials science and drug discovery. However, their utility is still limited by suboptimal performance on large molecular structures and limited training data. To address this gap, we explore the design space of E(3)-equivariant diffusion models, focusing on previously unexplored areas. Our extensive comparative analysis evaluates the interplay between continuous and discrete state spaces. From this investigation, we present the EQGAT-diff model, which consistently outperforms established models for the QM9 and GEOM-Drugs datasets. Significantly, EQGAT-diff takes continuous atom positions, while chemical elements and bond types are categorical and uses time-dependent loss weighting, substantially increasing training convergence, the quality of generated samples, and inference time. We also showcase that including chemically motivated additional features like hybridization states in the diffusion process enhances the validity of generated molecules. To further strengthen the applicability of diffusion models to limited training data, we investigate the transferability of EQGAT-diff trained on the large PubChem3D dataset with implicit hydrogen atoms to target different data distributions. Fine-tuning EQGAT-diff for just a few iterations shows an efficient distribution shift, further improving performance throughout data sets. Finally, we test our model on the Crossdocked data set for structure-based de novo ligand generation, underlining the importance of our findings showing state-of-the-art performance on Vina docking scores.

## 1 Introduction

The enormous success of machine learning (ML) in computer vision and natural language processing in recent years has led to the adaptation of ML in many research areas in the natural sciences, such as physics, chemistry, and biology, with promising results. Specifically, modern drug discovery widely utilizes ML to efficiently screen the vast chemical space for *de novo* molecule design in the early-stage drug discovery pipeline. An important aspect is the structure-based or target-aware design of novel molecules in 3D space (Schneuing et al., 2023; Guan et al., 2023; Stärk et al., 2022; Corso et al., 2023). However, incorporating the 3D geometries of molecules for rational and structure-based drug design is challenging, and the development of ML models in this domain is anything but easy, as these models need to function with just a limited amount of data to learn physical rules in 3D space accurately. Fortunately, applying geometric deep learning to molecule generation has gained attention in the scientific community in recent years, paving the way for innovative approaches. These result in diffusion models quickly becoming state-of-the-art in this area due to their ability to effectively learn complex data distributions (Hoogeboom et al., 2022; Igashov et al., 2022; Schneuing et al., 2023; Vignac et al., 2023; Guan et al., 2023). While this has enabled researchers to develop generative models for molecular design that can sample novel molecules in 3D space, several drawbacks and open questions remain prevalent for practitioners. Molecule gener-

---

[*]Equal contribution. Correspondence to {`tuan.le, julian.cremer`}@pfizer.com

ative models are required to both generate realistic molecules in 3D space and preserve fundamental chemical rules, i.e., correct bonding and valencies. Various design decisions have to be taken into account that heavily impact the performance and complexity of those models. Hence, there is a high need to better understand the design space of diffusion models for molecular modeling. Moreover, the availability of molecular data is not as abundant, confronting ML models with relatively narrow and specific data distributions. That is, ML models are usually trained explicitly for each data set, which is unfavorable regarding the efficient use of training data and computing resources.

This work introduces the E(3)-equivariant graph attention denoising neural network EQGAT-diff. We systematically explore the design space of 3D equivariant diffusion models, including various parameterizations, loss weightings, data, and input feature modalities. Beyond that, we explore an efficient pre-training scheme on molecular data with implicit hydrogens. This enables a data- and time-efficient training and fine-tuning procedure leading to higher molecule stability. Our contributions are the following:

- We propose EQGAT-diff – a fast and accurate 3D molecular diffusion model that employs E(3)-equivariant graph attention. Our proposed model achieves SOTA results in shorter training time and with less trainable parameters than previous architectures.
- We systematically explore various design choices for 3D molecular diffusion models and provide a thorough ablation study across the popular benchmark sets QM9 and GEOM-Drugs. We propose a time-dependent loss weighting as a crucial component for fast training convergence, better inference speed, and sample quality.
- We demonstrate the transferability of an EQGAT-diff model pre-trained on the PubChem3D dataset to smaller but complex molecular datasets. After a short fine-tuning on the target distribution, we show that the model outperforms models trained from scratch on the target data by only training on subsets.
- We extend the diffusion process by modeling chemically motivated additional features and show a further significant increase in performance.

In summary, we found the following ingredients to be crucial: E(3)-equivariant graph attention, time-dependent loss weighting, unconditional pretraining on large databases comprising 3D conformers like PubChem3D, and adding chemical features like aromaticity and hybridization state as feature input to the denoising diffusion model.

## 2 RELATED WORK

Denoising diffusion probabilistic models (DDPM) (Sohl-Dickstein et al., 2015; Ho et al., 2020; Kingma et al., 2021; Song et al., 2021b) have achieved great success in various generation tasks due to their remarkable ability to model complicated distributions in the image and text processing community (Popov et al., 2021; Kong et al., 2021; Salimans & Ho, 2022; Rombach et al., 2022; Karras et al., 2022; Li et al., 2022; Kingma & Gao, 2023). Deep generative modeling in the life sciences has become a promising research area, e.g., conditional conformer generation based on the 2D molecular graph, in which (Mansimov et al., 2019; Simm & Hernandez-Lobato, 2020) leverage the idea of variational autoencoders, while recent work by (Xu et al., 2022; Jing et al., 2022) use DDPMs, to predict the 3D coordinates with the help of 3D equivariant graph neural networks. In the *de novo* setting, another line of research focuses on directly generating the atomic coordinates and elements, using either autoregressive models (Gebauer et al., 2019; 2022; Luo & Ji, 2022), where atomic elements are generated one by one sequentially, or neural learning algorithms based on continuous normalizing flows (Satorras et al., 2021) that are computationally expensive due to the integration of an ordinary differential equation, leading to limited performance and scalability on large molecular systems. Diffusion models offer efficient training by progressively applying Gaussian noise to transform a complex data distribution to approximately tractable Gaussian prior, intending to learn the reverse process. Hoogeboom et al. (2022) introduced E(3) equivariant diffusion model (EDM) for *de novo* molecule design that simultaneously learns atomic elements next to the coordinates while treating chemical elements as continuous variables to utilize the formalism of DDPM. Follow-up works leverage EDM and develop diffusion models for linker design (Igashov et al., 2022) or ligand-protein complex modeling (Schneuing et al., 2023). Another line of work leverages the formalism of stochastic differential equations (SDEs) (Song et al., 2021b) and

Schroedinger Bridges with extension to manifolds (De Bortoli et al., 2021; 2022) to generating 3D conformer of a fixed molecule into a protein pocket (Corso et al., 2023), while (Wu et al., 2022) modifies the forward diffusion process to incorporate physical priors.

## 3 BACKGROUND

**Problem Formulation and Notation** We investigate the generation of molecular structures in a *de novo* setting, where atomic coordinates, chemical elements, and the bond topology are sampled. A molecular structure is given by , where the vertices $V = (v_1, \ldots, v_N)$ refer to the $N$ atoms. Each vertex is a tuple $v_i = (r_i, h_i)$ comprised of the atomic coordinate in 3D space $r_i$ and chemical element $h_i$. The latter is one-hot encoded for $K$ elements, i.e., $h_i = (0, 0, \ldots, 1, 0)^\top$. The edges $E = (e_{ij})_{i,j=0}^N$ describe the connectivity of the molecule, where each edge feature can take five distinct values, namely the existence of no bond or a single-, double-, triple- or aromatic bond between atom $i$ and $j$. Additionally, we exclude self-loops in our data representation. We write node features as matrices $\mathbf{X} \in \mathbb{R}^{N \times 3}$ and $\mathbf{H} \in \{0, 1\}^{N \times K}$, while the bond topology is given by $\mathbf{E} \in \{0, 1\}^{N \times N \times 5}$. We aim to develop a probabilistic model that is invariant to the permutation of atoms of the same chemical element and roto-translation of coordinates in 3D space. That means, regardless of how atom indices in the node-feature matrix $\mathbf{H}$ are shuffled and coordinates $\mathbf{X}$ roto-translated, the probability for a molecular structure $\mathcal{X}$ remains unchanged.

### 3.1 DENOISING DIFFUSION PROBABILISTIC MODELS

Discrete-time diffusion models (Sohl-Dickstein et al., 2015; Ho et al., 2020) are latent variable generative models characterized by a forward and reverse Markov process over $T$ steps. Given a sample from the data distribution $x_0 \sim q(x_0)$, the forward process $q(x_{1:T}|x_0) = \prod_{t=1}^T q(x_t|x_{t-1})$ transforms it into a sequence of increasingly noisy latent variables $x_{1:T} = (x_1, x_2, \ldots, x_T)$ and $x_i \in \mathcal{X}$. The learnable reverse Markov process $p_\theta(x_{0:T}) = p(x_T) \prod_{t=1}^T p_\theta(x_{t-1}|x_t)$ is trained to gradually denoise the latent variables approaching the data distribution. Sohl-Dickstein et al. (2015) initially proposed a diffusion process for binary and continuous data, while the latter consists of Gaussian transition kernels. The learning process for discrete data has been introduced by Hoogeboom et al. (2021) and Austin et al. (2021), leveraging categorical transition kernels in the form of doubly stochastic matrices. Crucially, both forward processes define tractable distributions determined by a noise schedule $\{\beta t\}_{t=1}^T$, such that the reverse generative model can be trained efficiently. As molecular data consists of atoms, bonds, and 3D coordinates, recent work leverages a combination of Gaussian and categorical diffusion for 3D molecular generation (Peng et al., 2023; Vignac et al., 2023; Guan et al., 2023). A subtle property of tractable transition kernels is that the distribution of a noisy state conditioned on a data sample is also tractable, and for continuous or discrete data follows a multivariate normal or categorical distribution

$$q(\mathbf{x}_t|\mathbf{x}_0) = \mathcal{N}(\mathbf{x}_t|\sqrt{\bar{\alpha}_t}\mathbf{x}_0, (1 - \bar{\alpha}_t)\mathbf{I}) \quad \text{and} \quad q(\mathbf{c}_t|\mathbf{c}_0) = \mathcal{C}(\mathbf{c}_t|\bar{\alpha}_t\mathbf{c}_0 + (1 - \bar{\alpha}_t)\tilde{\mathbf{c}}), \quad (1)$$

where $\bar{\alpha}_t = \prod_{k=1}^t (1 - \beta_k) \in (0, 1)$, and $(1 - \bar{\alpha}_t)$ determine a *variance-preserving* (VP) noise scheduler Song et al. (2021b). The vector $\tilde{\mathbf{c}}$ with $\tilde{\mathbf{c}}^\top \mathbf{1}_K = 1$ determines the prior distribution of the categorical diffusion, as $\bar{\alpha}_T \to 0$. Possible prior distributions are the uniform distribution over $K$-classes or the empirical distribution of categories in a dataset. In this work, we perturb atomic coordinates $\mathbf{X}$, chemical elements $\mathbf{H}$, and edge features $\mathbf{E}$ independently, using Gaussian and categorical diffusion. To conserve the edge-symmetry between atoms $i$ and $j$, we only perturb the upper-triangular elements of $\mathbf{E}$. Diffusion models are trained by maximizing the variational lower-bound of the data log-likelihood (Sohl-Dickstein et al., 2015; Kingma et al., 2021; Austin et al., 2021) decomposed as $\log p(x) \geq L_0 + L_{\text{prior}} + \sum_{t=1}^{T-1} L_t$, where $L_0 = \log p(x_0|x_1)$ and $L_{\text{prior}} = -D_{KL}(q(x_T|p(x_T))$ denote the reconstruction, and prior loss. These two loss terms are commonly neglected during optimization, while the diffusion loss $L_t = -D_{KL}[q(x_{t-1}|x_t, x_0)|p_\theta(x_{t-1}|x_t)]$ has a closed-form expression since $q(x_{t-1}|x_t, x_0)$ is either a multivariate normal or categorical distribution, enabling efficient KL divergence minimization by predicting the corresponding distribution parameters. These are defined as a function of $x_t$ and $x_0$, implying that the diffusion model is tasked to predict the clean data sample $\hat{x}_0$ to optimize $L_t$ (Ho et al., 2020; Austin et al., 2021).

## 4   EQGAT-DIFF

An essential requirement to obtain a data-efficient model is to reflect the permutational symmetry of atoms of the same chemical element and the roto-translational symmetries of 3D molecular structures. In machine learning force fields, it has been shown that rotationally invariant features alone do not accurately represent the 3D molecular structure and hence require higher-order equivariant features (Schütt et al., 2021; Batzner et al., 2022; Thölke & Fabritiis, 2022; Batatia et al., 2022).

In short, a function $f : \mathcal{X} \rightarrow \mathcal{Y}$ mapping from input space $\mathcal{X}$ to output space $\mathcal{Y}$ is equivariant to the group $G$ iff $f(g.x) = g.f(x)$, where $g.$ denotes the action of the group element $g \in G$ on an object $x, y \in \mathcal{X}, \mathcal{Y}$. As graph neural networks operate on graphs and map nodes into a feature space through shared transformations among all nodes, permutation equivariance is naturally preserved Bronstein et al. (2021). In contrast, point clouds are embedded in 3D space, so we additionally consider the rotation, reflection, and translation group in $\mathbb{R}^3$, often abbreviated as E(3). For the atomic coordinates, we require that $f(\mathbf{X}\mathbf{Q} + \mathbf{t}) = f(\mathbf{X})\mathbf{Q} + \mathbf{t}$, where $\mathbf{Q} \in O(3)$ is a rotation or reflection matrix and $\mathbf{t} \in \mathbb{R}^3$ a translation vector added row-wise. Group equivariance of a function $f$ in the context of a diffusion model for molecular data is a requirement to preserve the group invariance for a probability density, as shown by Köhler et al. (2020) and Xu et al. (2022). To better address the challenge of molecular modeling, we propose a modified version of the EQGAT architecture Le et al. (2022), coined EQGAT-diff, which leverages attention-based feature aggregation of neighboring nodes. EQGAT-diff employs rotation equivariant vector features that can be interpreted as learnable vector bundles, which the denoising networks of EDM Hoogeboom et al. (2022) and MiDi Vignac et al. (2023) are lacking. Point clouds are modeled as fully connected graphs, so message passing computes all pairwise interactions. Equivariant vector features are obtained through a tensor product of scalar features with normalized relative positions $\mathbf{x}_{(ji,n)} = \frac{1}{||\mathbf{x_j} - \mathbf{x_i}||}(\mathbf{x}_j - \mathbf{x}_i)$ as similarly proposed in the works of Jing et al. (2021) and Schütt et al. (2021). We iteratively update hidden edge features within the EQGAT-diff architecture to handle the edge prediction between two atoms. To achieve this, we modify the message function of EQGAT as

$$\mathbf{m}_{ji}^{(l)} = \mathrm{MLP}([\mathbf{h}_j^{(l)}; \mathbf{h}_i^{(l)}; \mathbf{W}_{e_0}^{(l)}\mathbf{e}_{ji}^{(l)}; d_{ji}^{(l)}; d_j^{(l)}; d_i^{(l)}; \mathbf{p}_j^{(l)} \cdot \mathbf{p}_i^{(l)}]),$$

where ; denotes concatenation of E(3) invariant embeddings and MLP is a 2-layer multi-layer perceptron. The message embedding $\mathbf{m}_{ji}^{(l)} = (\mathbf{a}_{ji}^{(l)}, \mathbf{b}_{ji}^{(l)}, \mathbf{c}_{ji}^{(l)}, \mathbf{d}_{ji}^{(l)}, s_{ji}^{(l)})^\top \in \mathbb{R}^K$ is further split into sub-embeddings that serve as filter to aggregate node information from all other source nodes $j$.

$$\mathbf{h}_i^{(l+1)} = \mathbf{h}_i^{(l)} + \sum_j \frac{\exp(\mathbf{a}_{ji}^{(l)})}{\sum_{j'} \exp(\mathbf{a}_{j'i}^{(l)})} \mathbf{W}_h^{(l)} \mathbf{h}_j^{(l)} \quad \text{and} \quad \mathbf{e}_{ji}^{(l+1)} = \mathbf{W}_{e_1}^{(l)} \sigma(\mathbf{e}_{ji}^{(l)} + \mathbf{d}_{ji}^{(l)}),$$

$$\mathbf{v}_i^{(l+1)} = \mathbf{v}_i^{(l)} + \frac{1}{N} \sum_j \mathbf{x}_{ji,n} \otimes \mathbf{b}_{ji}^{(l)} + (\mathbf{1} \otimes \mathbf{c}_{ji}^{(l)}) \odot \mathbf{v}_j^{(l+1)} \mathbf{W}_v^{(l)},$$

$$\mathbf{x}_i^{(l+1)} = \mathbf{x}_i^{(l)} + \frac{1}{N} \sum_j s_{ji}^{(l)} \mathbf{x}_{ji,n}^{(l)},$$

where $\mathbf{1} = (1, 1, 1)^\top$ and $\sigma$ is the SiLU activation function. The embeddings are further updated and normalized with details explained in the Appendix A.1.

## 5   EXPLORING THE DESIGN SPACE OF 3D MOLECULAR DIFFUSION MODELS

The design space of diffusion models has many degrees of freedom concerning, among others, the data representation, training objective, forward inference process, and the denoising neural network. In *de novo* 3D molecular generation, Hoogeboom et al. (2022) (EDM) utilized the $\epsilon$-parameterization and proposed to model chemical elements as well as atomic positions continuously. Vignac et al. (2023) proposed MiDi, which generates the molecular graph and 3D structure simultaneously. This model uses the $x_0$-parameterization and employs the framework developed by Austin et al. (2021) to model not only chemical elements but also formal charges and bond types in discrete state space. Both parameterizations optimize the same objective, i.e., aiming to minimize the KL divergence $D_{KL}[q(x_{t-1}|x_t, x_0)|p_\theta(x_{t-1}|x_t)]$. Ho et al. (2020) found that optimizing the diffusion model in noise-space on images results in improved generation performance than predicting

Table 1: Comparison of EQGAT-diff on QM9 and GEOM-Drugs trained with $w_u$ or $w_s(t)$ loss-weighting. We report the mean values over five runs of selected evaluation metrics with the margin of error for the 95% confidence level given as subscripts. The best results are in bold.

| | QM9 | | | GEOM-Drugs | | |
|---|---|---|---|---|---|---|
| Weighting | Mol. Stability ↑ | Validity ↑ | Connect. Comp. ↑ | Mol. Stability ↑ | Validity ↑ | Connect. Comp. ↑ |
| $w_u$ | $97.39_{\pm 0.23}$ | $97.99_{\pm 0.20}$ | $99.70_{\pm 0.03}$ | $87.59_{\pm 0.19}$ | $71.44_{\pm 0.22}$ | $86.57_{\pm 0.33}$ |
| $w_s(t)$ | $\mathbf{98.68}_{\pm 0.11}$ | $\mathbf{98.96}_{\pm 0.07}$ | $\mathbf{99.94}_{\pm 0.03}$ | $\mathbf{91.60}_{\pm 0.14}$ | $\mathbf{84.02}_{\pm 0.19}$ | $\mathbf{95.08}_{\pm 0.12}$ |

the original image from a noised version. While noise prediction might benefit the image domain, this does not necessarily generalize to 3D molecular data. In fact, MiDi outperforms EDM across all standard benchmark metrics and datasets. However, whether the improved performance stems from the $x_0$-parameterization, the employment of categorical diffusion for discrete features, or using bond types and other chemical features has still been unclear, leaving researchers and practitioners guessing which kind of diffusion model to deploy in their respective tasks.

In this section, we explore the design space of *de novo* molecular diffusion models in these three aspects while consistently using EQGAT-diff as the denoising neural network to isolate the effect of each change for better comparison. The diffusion models are evaluated on the QM9 dataset (Ramakrishnan et al., 2014) containing molecules with up to 9 heavy atoms, and the GEOM-Drugs dataset (Axelrod & Gómez-Bombarelli, 2022) containing up to 15 heavy atoms. We utilize the data splits from Vignac et al. (2023) and benchmark all models on full molecular 3D graphs that include explicit hydrogens.

## 5.1 TRAINING DETAILS

We either employ noise prediction ($\epsilon$-parameterization) or data prediction ($x_0$-parameterization) to train EQGAT-diff , such that the group equivariant network $f_\theta(x_t)$ receives a noisy molecule $x_t = (\mathbf{X}_t, \mathbf{H}_t, \mathbf{E}_t)$ and either outputs the applied noise $\hat{\epsilon}_t = (\hat{\epsilon}_{\mathbf{X}_t}, \hat{\epsilon}_{\mathbf{H}_t}, \hat{\epsilon}_{\mathbf{E}_t})$ or a prediction of the clean data $\hat{x}_0 = (\hat{\mathbf{X}}_0, \hat{\mathbf{H}}_0, \hat{\mathbf{E}}_0)$ of coordinates, chemical elements as well as bonds. We draw a random batch of molecules and uniformly sample steps $t \in \mathcal{U}(1, T)$ and optimize the diffusion loss $L_t$ for each sample. While we use the mean squared error loss for the $\epsilon$-model, the $x_0$-model is optimized using loss functions $l_d$ depending on the data modality $d$. Here, $l_d$ is a mean squared error for continuous and the cross-entropy loss for categorical data. This leads to a composite loss

$$L_{t,\epsilon} = w(t)||\epsilon_t - \hat{\epsilon}_\theta(x_t, t)||^2 \quad \text{and} \quad L_{t,x_0} = w(t) \cdot l_d(x_0, \hat{x}_\theta(x_t, t); \lambda_m), \quad (2)$$

where $\lambda_m$ denotes a modality-dependent weighting, which we adopt from Vignac et al. (2023) and set to $\lambda_x = 3, \lambda_h = 0.4, \lambda_e = 2$. For noise learning, we adopt an atom-type feature scaling of 0.25 as in Hoogeboom et al. (2022). Notably, $w(t)$ is a loss weighting commonly set to 1 across all time steps, which has been previously found to work best (Ho et al., 2020). In contrast to this result, we find this term to be crucial for molecular design, as discussed in Sec. A.4. Following Vignac et al. (2023), we also employ an adaptive noise schedule (see Appendix A.1.1).

## 5.2 METRICS

Following (Hoogeboom et al., 2022), we measure validity using the success rate of RDKit sanitization over 10,000 molecules (pre-selecting connected components only) - with the caveat that the RDKit sanitization might add implicit hydrogens to the system to satisfy the chemical constraints. Therefore, checking atomic and molecular stability for the correct valencies using a pre-defined lookup table that complements the validation is essential. Further, we propose to include diversity/similarity measures. We evaluate the diversity of sampled molecules using the average Tanimoto distance and measure the similarity with the training dataset via Kullback-Leibler divergence and the Tanimoto distance. Lastly, following Vignac et al. (2023), we use the atom and bond total variations (AtomsTV and BondsTV) that measure the $l_1$ distance between the marginal distribution of atom types and bond types for the generated set and the test set, respectively. Moreover, we employ the Wasserstein distance between valencies, bond lengths, and bond angles, with the latter two being 3D metrics to evaluate conformer accuracy. For more details, we refer to Vignac et al. (2023) and Appendix A.2.

Kingma et al. (2021) have shown that the intermediate KL-divergence loss $L_t$ in the variational lower bound (VLB) for a Gaussian diffusion can be simplified to

$$L_t = \frac{1}{2}(w(t))||x_0 - x_\theta(x_t,t)||_2^2 = \frac{1}{2}\mathbb{E}_{\epsilon \sim \mathcal{N}(0,I)}[(\text{SNR}(t-1) - \text{SNR}(t))||x_0 - x_\theta(x_t,t)||_2^2],$$

where $\text{SNR}(t) = \frac{\bar{\alpha}_t}{1-\bar{\alpha}_t}$ refers to the signal-to-noise ratio. However, the weighting coefficients in diffusion models are commonly set to 1, i.e., $w_u = 1$ in EDM or MiDi (Hoogeboom et al., 2022; Vignac et al., 2023).

We hypothesize that denoising requires high accuracy for timesteps close to the data distribution to generate valid molecules, while errors close to the noise distribution are neglectable. Such loss weighting has been proposed by Salimans & Ho (2022) as 'truncated SNR', which we modify for our use case. Specifically, we perform experiments with the loss weighting

$$w_s(t) = \max(0.05, \ \min(1.5, \ \text{SNR}(t))), \tag{3}$$

which matches our hypothesis about learning with higher weightings approaching the data distribution (see A.4.1 and Fig. 5). We clip the maximum value of $1.5$ to enforce larger weightings to enhance learning compared to uniform weighting, followed by an abrupt exponential decay. We train EQGAT-diff using Gaussian diffusion on atomic coordinates and categorical diffusion for chemical elements, formal charges, and bond features following the parameterization proposed by Vignac et al. (2023), predicting a clean data sample $\hat{x}_0$ given a noisy version $x_t$. As shown in Table 1, training EQGAT-diff on GEOM-Drugs with $w_s(t)$ results in a better generative model that can sample molecules preserving chemistry rules, measured in increased molecule stability of 91.60%, compared to the EQGAT-diff which was trained with $w_u$, only achieving 87.59%. As the $w_s(t)$ loss weighting achieved better evaluation metrics and significantly faster training convergence on the QM9 and GEOM-Drugs datasets, we choose it as default for the following experiments conducted in this work. We provide further empirical evidence in Appendix A.3

Table 2: Overall performance of EQGAT-diff on QM9 and GEOM-Drugs for discrete and continuous diffusion as well as noise ($\epsilon$) and data learning ($x_0$). Discrete or continuous diffusion is denoted as 'disc' and 'cont', respectively, given as subscripts, $\epsilon$- and $x_0$-parameterization as superscripts. We report mean values over five sampling runs with 95% confidence intervals as subscripts. The best results are in bold.

| Dataset | QM9 | | | GEOM-Drugs | | |
|---|---|---|---|---|---|---|
| Model | $\text{EQGAT}_{disc}^{x_0}$ | $\text{EQGAT}_{cont}^{x_0}$ | $\text{EQGAT}_{cont}^{\epsilon}$ | $\text{EQGAT}_{disc}^{x_0}$ | $\text{EQGAT}_{cont}^{x_0}$ | $\text{EQGAT}_{cont}^{\epsilon}$ |
| Mol. Stab. ↑ | $\mathbf{98.68}_{\pm 0.11}$ | $96.45_{\pm 0.17}$ | $96.18_{\pm 0.16}$ | $\mathbf{91.60}_{\pm 0.14}$ | $90.46_{\pm 0.09}$ | $85.19_{\pm 0.72}$ |
| Atom. Stab ↑ | $\mathbf{99.92}_{\pm 0.00}$ | $99.79_{\pm 0.01}$ | $99.68_{\pm 0.02}$ | $\mathbf{99.72}_{\pm 0.01}$ | $\mathbf{99.73}_{\pm 0.01}$ | $99.32_{\pm 0.04}$ |
| Validity ↑ | $\mathbf{98.96}_{\pm 0.07}$ | $96.79_{\pm 0.15}$ | $97.04_{\pm 0.17}$ | $\mathbf{84.02}_{\pm 0.19}$ | $80.96_{\pm 0.38}$ | $79.13_{\pm 0.58}$ |
| Connect. Comp. ↑ | $\mathbf{99.94}_{\pm 0.03}$ | $99.82_{\pm 0.05}$ | $99.71_{\pm 0.03}$ | $\mathbf{95.08}_{\pm 0.12}$ | $93.30_{\pm 0.21}$ | $94.10_{\pm 0.48}$ |
| Novelty ↑ | $64.03_{\pm 0.24}$ | $60.96_{\pm 0.54}$ | $73.40_{\pm 0.32}$ | $\mathbf{99.87}_{\pm 0.04}$ | $\mathbf{99.83}_{\pm 0.04}$ | $99.82_{\pm 0.0}$ |
| Uniqueness ↑ | $\mathbf{100.00}_{\pm 0.00}$ | $\mathbf{100.0}_{\pm 0.00}$ | $\mathbf{100.00}_{\pm 0.00}$ | $\mathbf{100.00}_{\pm 0.00}$ | $\mathbf{100.00}_{\pm 0.00}$ | $\mathbf{100.00}_{\pm 0.00}$ |
| Diversity ↑ | $91.72_{\pm 0.02}$ | $91.51_{\pm 0.03}$ | $\mathbf{91.89}_{\pm 0.03}$ | $\mathbf{89.00}_{\pm 0.03}$ | $88.87_{\pm 0.04}$ | $88.97_{\pm 0.05}$ |
| KL Divergence ↑ | $91.36_{\pm 0.29}$ | $\mathbf{91.41}_{\pm 0.54}$ | $88.97_{\pm 0.31}$ | $87.17_{\pm 0.34}$ | $87.35_{\pm 0.35}$ | $\mathbf{87.70}_{\pm 0.58}$ |
| Train Similarity ↓ | $0.076_{\pm 0.00}$ | $0.076_{\pm 0.00}$ | $\mathbf{0.075}_{\pm 0.00}$ | $\mathbf{0.113}_{\pm 0.00}$ | $0.114_{\pm 0.00}$ | $0.114_{\pm 0.00}$ |
| AtomsTV [$10^{-2}$] ↓ | $1.0_{\pm 0.00}$ | $2.0_{\pm 0.00}$ | $2.7_{\pm 0.00}$ | $3.4_{\pm 0.10}$ | $3.6_{\pm 0.10}$ | $\mathbf{2.9}_{\pm 0.20}$ |
| BondsTV [$10^{-2}$] ↓ | $1.2_{\pm 0.00}$ | $1.8_{\pm 0.00}$ | $1.2_{\pm 0.00}$ | $\mathbf{2.4}_{\pm 0.00}$ | $\mathbf{2.4}_{\pm 0.00}$ | $\mathbf{2.4}_{\pm 0.00}$ |
| ValencyW$_1$ [$10^{-2}$] ↓ | $0.6_{\pm 0.10}$ | $1.9_{\pm 0.00}$ | $0.9_{\pm 0.00}$ | $1.2_{\pm 0.10}$ | $1.9_{\pm 0.10}$ | $1.6_{\pm 0.00}$ |
| BondLenghtsW$_1$ [$10^{-2}$] ↓ | $\mathbf{0.2}_{\pm 0.10}$ | $0.5_{\pm 0.00}$ | $\mathbf{0.2}_{\pm 0.10}$ | $\mathbf{0.2}_{\pm 0.10}$ | $0.3_{\pm 0.00}$ | $0.7_{\pm 0.40}$ |
| BondAnglesW$_1$ ↓ | $\mathbf{0.42}_{\pm 0.03}$ | $1.86_{\pm 0.06}$ | $0.52_{\pm 0.03}$ | $\mathbf{0.92}_{\pm 0.02}$ | $0.95_{\pm 0.02}$ | $1.07_{\pm 0.06}$ |

## 5.3 DIFFUSION PARAMETERIZATION: $\epsilon$ VS $x_0$ AND DISCRETE VS CONTINUOUS

Diffusion models for continuous data are commonly implemented using the $\epsilon$-parameterization Ho et al. (2020), which is connected to denoising score matching models proposed by Song & Ermon (2019). Diffusion models have quickly adapted this setting for 3D molecular design (Hoogeboom et al., 2022; Igashov et al., 2022; Schneuing et al., 2023). However, no comparative study of $x_0$ and $\epsilon$-parameterization in this domain has been performed yet. To close this gap, we benchmark the $\epsilon$- vs. the $x_0$-parameterization on data modalities subject to a Gaussian diffusion. That is, we treat all node features (including atomic elements, charges, and coordinates) as well as the bond features as continuous variables and optimize our diffusion model using either the $\epsilon$- or $x_0$-parameterization with the loss functions defined in Eq. (2).

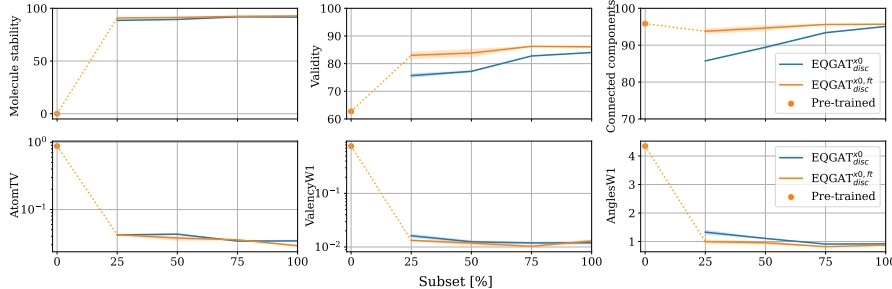

Figure 1: Selected evaluation metrics for EQGAT-diff trained on GEOM-Drugs subsets (25, 50, 75%) from scratch or fine-tuned. We also report the results of the pre-trained, not fine-tuned model (0%).

In the following, we abbreviate EQGAT-diff with EQGAT to keep the notation clear, depicting the diffusion type subscripted and the parameterization superscripted. Table 2 shows that the $x_0$-parameterization ($\text{EQGAT}^{x_0}_{cont}$) achieves higher molecule stability on QM9 and GEOM-Drugs than the $\epsilon$-parameterization ($\text{EQGAT}^{\epsilon}_{cont}$). The performance gap is pronounced on the GEOM-Drugs dataset, which covers a broader range of larger and more complex molecules. On this more demanding benchmark, $\text{EQGAT}^{x_0}_{cont}$ outperforms the $\epsilon$-model with $90.46\%$ molecule stability against $85.19\%$. The lower molecule stability for the $\epsilon$-model is due to the molecular graph not being accurately denoised during the sampling. Thus, the final edge features do not preserve the valency constraints of the chemical elements.

Next, we compare how the choice of categorical or Gaussian diffusion for modeling the chemical elements, charges, and edge features affects the generation performance. Recall that the noising process in the categorical diffusion perturbs the one-hot encoding of discrete features by jumping from one class to another, or staying on the same class. Alternatively, noise from a multivariate normal distribution is added to the (scaled) one-hot encodings, as described in Eq. (1). For both settings, the diffusion models ($\text{EQGAT}^{x_0}_{disc}$ and $\text{EQGAT}^{x_0}_{cont}$) are tasked with predicting the original data point $x_0$, as there is no $\epsilon$-parameterization when employing categorical diffusion. The previous ablation has shown that data prediction is superior to noise prediction when dealing with molecular data in a continuous setting. We discover that $\text{EQGAT}^{x_0}_{disc}$ outperforms $\text{EQGAT}^{x_0}_{cont}$ in all evaluation metrics on the QM9 and GEOM-Drugs dataset as shown in Table 2. Hence, employing the categorical diffusion for discrete state-space in the $x_0$-parameterization is the preferred choice.

## 6 TRANSFERABILITY OF MOLECULAR DIFFUSION MODELS

In many molecular design scenarios, only a limited amount of training data is available for a desired target distribution, e.g., in structure-based drug design. However, 3D generative molecular diffusion models require a lot of training data to yield a high ratio of valid and novel molecules. This section investigates how well a diffusion model pre-trained on a general large set of molecules transfers to a target distribution specified by a small training set of complex molecular structures. We use the PubChem3D dataset Bolton et al. (2011) for pre-training, which consists of roughly 95.7 million compounds from the PubChem database. It includes all molecules with chemical elements H, C, N, O, F, Si, P, S, Cl, Br, and I with less than $50$ non-hydrogen atoms and a maximum of $15$ rotatable bonds. The 3D structures have been computed using OpenEye's OMEGA software (Hawkins & Nicholls, 2012). We train EQGAT-diff on PubChem3D on four Nvidia A100 GPUs for one epoch ($\sim 24$ hours). Interestingly, we found that by reducing the size of molecular graphs using only implicit hydrogens, we could reduce the

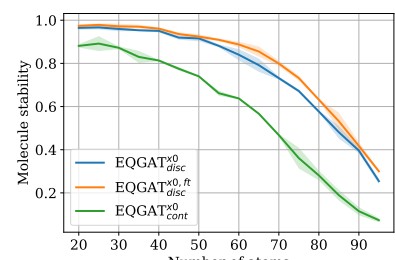

Figure 2: Comparing $\text{EQGAT}^{x_0,ft}_{disc}$ with $\text{EQGAT}^{x_0}_{disc}$, and $\text{EQGAT}^{x_0}_{cont}$ regarding molecule stability of 600 generated molecules with an increasing number of atoms. Standard deviations are plotted in shaded areas.

Table 3: Comparison of EQGAT$_{disc}$ models trained for 800 epochs on GEOM-Drugs. The superscripts 'ft' and 'af' abbreviate *fine-tuned* and *additional-features*. The margin of error for the 95% confidence level is given as subscripts. We also compare EDM and the current SOTA, MiDi. Training details for MiDi are given in Appendix A.6. The best results are in bold.

| Dataset | GEOM-Drugs | | | | | |
|---|---|---|---|---|---|---|
| **Model** | EQGAT$_{disc}^{x0}$ | EQGAT$_{disc}^{x0,ft}$ | EQGAT$_{disc}^{x0,af}$ | EQGAT$_{disc}^{x0,af,ft}$ | EDM | MiDi |
| Mol. Stab. ↑ | $93.11_{\pm0.31}$ | $93.92_{\pm0.13}$ | $\mathbf{94.51}_{\pm0.18}$ | $\mathbf{95.01}_{\pm0.37}$ | 40.3 | $89.7_{\pm0.60}$ |
| Atom. Stab ↑ | $99.79_{\pm0.01}$ | $\mathbf{99.81}_{\pm0.01}$ | $\mathbf{99.83}_{\pm0.01}$ | $\mathbf{99.84}_{\pm0.00}$ | 97.8 | $99.7_{\pm0.01}$ |
| Validity ↑ | $85.86_{\pm0.33}$ | $\mathbf{88.04}_{\pm0.17}$ | $87.89_{\pm0.31}$ | $\mathbf{88.42}_{\pm0.26}$ | 87.8 | $70.5_{\pm0.41}$ |
| Connect. Comp. ↑ | $\mathbf{96.32}_{\pm0.25}$ | $\mathbf{96.57}_{\pm0.18}$ | $96.36_{\pm0.25}$ | $\mathbf{96.71}_{\pm0.20}$ | 41.4 | $88.76_{\pm0.55}$ |
| Novelty ↑ | $99.82_{\pm0.05}$ | $99.84_{\pm0.02}$ | $99.82_{\pm0.05}$ | $99.82_{\pm0.03}$ | **100.00** | $\mathbf{100.00}_{\pm0.00}$ |
| Diversity ↑ | $\mathbf{89.03}_{\pm0.03}$ | $\mathbf{89.05}_{\pm0.05}$ | $88.98_{\pm0.02}$ | $88.96_{\pm0.01}$ | - | - |
| KL Divergence ↑ | $87.66_{\pm0.31}$ | $87.58_{\pm0.56}$ | $\mathbf{88.38}_{\pm0.25}$ | $87.62_{\pm0.19}$ | - | - |
| Train Similarity ↓ | $0.114_{\pm0.0}$ | $\mathbf{0.113}_{\pm0.0}$ | $0.114_{\pm0.0}$ | $0.114_{\pm0.0}$ | - | |
| AtomsTV $[10^{-2}]$ ↓ | $3.02_{\pm0.08}$ | $3.02_{\pm0.10}$ | $\mathbf{2.88}_{\pm0.10}$ | $2.91_{\pm0.10}$ | 21.2 | $5.11_{\pm0.19}$ |
| BondsTV $[10^{-2}]$ ↓ | $2.44_{\pm0.01}$ | $\mathbf{2.40}_{\pm0.00}$ | $2.42_{\pm0.00}$ | $\mathbf{2.40}_{\pm0.00}$ | 4.8 | $2.44_{\pm0.00}$ |
| ValencyW$_1$ $[10^{-2}]$ ↓ | $1.18_{\pm0.09}$ | $1.20_{\pm0.00}$ | $\mathbf{0.85}_{\pm0.12}$ | $\mathbf{0.90}_{\pm0.10}$ | 28.5 | $2.48_{\pm0.52}$ |
| BondLenghtsW$_1$ $[10^{-2}]$ ↓ | $0.56_{\pm0.38}$ | $\mathbf{0.10}_{\pm0.00}$ | $0.50_{\pm0.51}$ | $0.20_{\pm0.10}$ | 0.2 | $0.2_{\pm0.10}$ |
| BondAnglesW$_1$ ↓ | $0.83_{\pm0.03}$ | $0.79_{\pm0.02}$ | $0.65_{\pm0.01}$ | $\mathbf{0.62}_{\pm0.01}$ | 6.23 | $1.73_{\pm0.32}$ |

pre-training time significantly without sacrificing performance in fine-tuning. For a comparison to keeping explicit hydrogens in the pre-training, see Appendix A.5. During fine-tuning, the diffusion model is tasked to adapt to the distribution of another dataset, now including explicit hydrogens.

To evaluate the effectiveness of pre-training, we fine-tune subsets of $(25, 50, 75\%)$ of the QM9 and GEOM-Drugs datasets. Our results suggest that using a pre-trained model and subsequent fine-tuning shows consistently superior performance across datasets, partly by a large margin (see Fig. 1). We demonstrate the importance of pre-training by evaluating molecule stability, validity, and the number of connected components of a fine-tuned model compared to training from scratch on the full data and its $25, 50, 75\%$ subsets. As a reference point $(0\%)$, we show the pre-trained model without fine-tuning evaluated on the aforementioned metrics. Interestingly, the fine-tuned model shares similar (best) scores with EQGAT$_{disc}^{x0}$ trained from scratch on 100% of the data when looking at atom type variation and valency as well as angle distance metrics using a hold-out test set as a reference. These metrics capture how well the model learns the underlying data distribution.

We find that the fine-tuned model effectively learns a distribution shift on GEOM-Drugs by only being trained on small subsets of the data. We list more detailed evaluation metrics and the evaluation on QM9 in Appendix A.3. Comparing the fine-tuned model EQGAT$_{disc}^{x0,ft}$ with EQGAT$_{disc}^{x0}$, and EQGAT$_{cont}^{x0}$, respectively, shown in Fig. 2, we can also observe that the fine-tuning leads to significantly more stable predictions for larger molecules. We suspect that these findings might also apply to learning building blocks on large databases like the Enamine REAL Space to bias the generative model towards, e.g., higher synthesizability while ensuring an efficient distribution shift on the target distribution.

## 7 INSERTING CHEMICAL DOMAIN KNOWLEDGE

In the previous sections, we examined and outlined the importance of design choices when employing diffusion models for 3D molecular generation. Taking these results, we select the best two models - with and without fine-tuning: EQGAT$_{disc}^{x0,ft}$ and EQGAT$_{disc}^{x0}$ - and train them to full convergence, comparing with EDM and MiDi. We demonstrate in Tab. 3 that EQGAT$_{disc}^{x0,ft}$, and even more so EQGAT$_{disc}^{x0,af}$ and EQGAT$_{disc}^{x0,af,ft}$, outperform MiDi on all evaluation metrics by a large margin, while, most notably, our models converge significantly faster and are twice as fast computationally (see Appendix A.6). Given the demonstrated ability of diffusion models to learn the data distribution of complex molecular structures, we insert more chemical domain knowledge into the diffusion model, going beyond bonding. We additionally utilize aromaticity, ring correspondence, and hybridization states to provide a more comprehensive description of the molecular structure. The new additional features are independently perturbed using the categorical transition kernels (see Eq. (1)) and subsequently denoised by our model. We observe that these additional chemical features again improve the performance of our models (EQGAT$_{disc}^{x0,af}$ and EQGAT$_{disc}^{x0,af,ft}$) compared to our previous models as well as EDM and MiDi.

## 8 STRUCTURE-BASED DE NOVO LIGAND DESIGN

We train EQGAT-diff on the Crossdocked dataset Francoeur et al. (2020) for de novo structure-based ligand design. Following (Guan et al., 2023) and (Schneuing et al., 2023), we consider the protein pocket as a condition to generate novel ligands. Here, the pocket is seen as a fixed 3D context, while the ligand's coordinates, atom and bond types get diffused and denoised. In Tab. 4, we report the validity, number of connected components as well as the Wasserstein distances of bond lengths and angles between generated set to the training set, respectively. We observe that the finetuned model with timestep loss weighting significantly outperforms the models that are trained from scratch on all metrics. For the models trained from scratch, using timestep weighting shows better performance than no loss weighting. These results further underline the relevance of our findings allowing for an effective transfer of our model to structure-based molecule generation.

Table 4: Comparison of EQGAT-diff models trained on the Crossdocked dataset for pocket-conditioned de novo ligand generation. $EQGAT_{disc}^{x0}$ and $EQGAT_{disc}^{x0,ft}$ are compared with and without loss weighting, each trained for 300 epochs. Mean values are reported over five runs of selected evaluation metrics with the margin of error for the 95% confidence level given as subscripts and best results in bold.

| Model | Validity ↑ | Connect. Comp. ↑ | BondLengths W1 $[10^{-2}]$ ↓ | BondAngles W1 ↓ |
|---|---|---|---|---|
| $EQGAT_{disc}^{x0}(w_u)$ | $85.51_{\pm 0.09}$ | $95.15_{\pm 0.14}$ | $0.20_{\pm 0.0}$ | $4.37_{\pm 0.20}$ |
| $EQGAT_{disc}^{x0}(w_s(t))$ | $89.62_{\pm 0.08}$ | $97.65_{\pm 0.11}$ | $0.12_{\pm 0.0}$ | $2.12_{\pm 0.26}$ |
| $EQGAT_{disc}^{x0,ft}(w_s(t))$ | $\mathbf{95.65}_{\pm 0.12}$ | $\mathbf{99.66}_{\pm 0.10}$ | $\mathbf{0.11}_{\pm 0.0}$ | $\mathbf{1.55}_{\pm 0.21}$ |

Based on these results, we sample ligands from $EQGAT_{disc}^{x0,ft}$ for docking. Following Luo et al. (2021), Peng et al. (2022), we draw 100 valid ligands per protein pocket and evaluate them using Vina (Hassan et al., 2017) as an empirical proxy of the ligand binding affinity. As shown in Tab. 5, $EQGAT_{disc}^{x0,ft}$ outperforms both TargetDiff Guan et al. (2023) and DiffSBDD Schneuing et al. (2023) on the docking score and across all other metrics while generating more diverse ligands.

Table 5: Docking performance comparison between $EQGAT_{disc}^{x0,ft}$, TargetDiff and DiffSBDD trained on the Crossdocked dataset for pocket-conditioned de novo ligand generation. Best results in bold.

| Model | Vina (All) ↓ | Vina (Top-10%) ↓ | QED ↑ | SA ↑ | Lipinski ↑ | Diversity ↑ |
|---|---|---|---|---|---|---|
| $EQGAT_{disc}^{x0,ft}(w_s(t))$ | $\mathbf{-7.423}_{\pm 2.33}$ | $-9.571_{\pm 2.14}$ | $\mathbf{0.522}_{\pm 0.18}$ | $\mathbf{0.697}_{\pm 0.20}$ | $4.66_{\pm 0.72}$ | $\mathbf{0.742}_{\pm 0.07}$ |
| TargetDiff | $-7.318_{\pm 2.47}$ | $\mathbf{-9.669}_{\pm 2.55}$ | $0.483_{\pm 0.20}$ | $0.584_{\pm 0.13}$ | $4.594_{\pm 0.83}$ | $0.718_{\pm 0.09}$ |
| DiffSBDD-cond | $-6.950_{\pm 2.06}$ | $-9.120_{\pm 2.16}$ | $0.469_{\pm 0.21}$ | $0.578_{\pm 0.13}$ | $4.562_{\pm 0.89}$ | $0.728_{\pm 0.07}$ |

## 9 CONCLUSIONS

In this work, we have introduced EQGAT-diff, a framework for fast and accurate end-to-end differentiable *de novo* molecule generation in 3D space, jointly predicting geometry, topology, chemical composition and optionally other chemical features like the hybridization. The findings presented here are underpinned by comprehensive ablation studies, which address a previously scientific blank spot by thoroughly exploring the design space of 3D equivariant diffusion models. We have specifically designed an equivariant diffusion model that combines Gaussian and discrete state space diffusion. Crucially, we have incorporated a timestep-dependent loss weighting that significantly enhances the performance and training time of EQGAT-diff and, furthermore, showcased the transferability of our model being pre-trained on PubChem3D on small datasets. Our proposed models have significantly surpassed the current state-of-the-art 3D diffusion models, particularly in generating larger and more complex molecules, as evidenced by their high molecule stability and validity, which evaluate that chemistry rules are preserved. Most notably, we also showcased that our framework seamlessly transfers to target-conditioned de novo ligand design superior docking scores while ensuring high diversity in samples. Given these achievements, we anticipate our findings will open avenues for ML-driven *de novo* structure-based drug discovery.

## CODE AVAILABILITY

Our source code and implementation will be released under `https://github.com/pfizer-opensource/eqgat-diff`.

## ACKNOWLEDGMENTS

This study was partially funded by the European Union's Horizon 2020 research and innovation program under the Marie Skłodowska-Curie Innovative Training Network European Industrial Doctorate grant agreement No. 956832 "Advanced machine learning for Innovative Drug Discovery."

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

# A APPENDIX

## A.1 MODEL DETAILS

Before message passing, we create a time embedding $t_e = \frac{t}{T} = \frac{t}{500}$ and concatenate those to the geometric-invariant (scalar) features, including atomic elements and charges, to pass the timestep information into the network. After each round of message passing, we employ a normalization layer for the position updates as proposed by Vignac et al. (2023), while scalar and vector features $(\mathbf{h}, \mathbf{v})$ are normalized using a Layernorm followed by an update block using gated equivariant transformation as proposed in the original EQGAT architecture (Le et al., 2022). After $L$ round of message passing and update blocks, we leverage the last layers' embeddings to perform the final prediction $\hat{x}_0 = (\hat{\mathbf{X}}, \hat{\mathbf{H}}, \hat{\mathbf{E}})$ as shown in Figure 3. For the case that additional (geometric) invariant features are modeled, including the atomic formal charges, aromaticity, or hybridization state, the hidden node matrix $\hat{\mathbf{H}}$ includes them as output prediction by simple concatenation, i.e., predicting more output channels.

We implement EQGAT-diff using PyTorch Geometric (Fey & Lenssen, 2019) and leverage the (sparse) coordinate (COO) format that stores the molecular data and respective edge indices of the fully connected graphs.

### A.1.1 MODEL TRAINING

We optimize EQGAT-diff under $x_0$ parameterization utilizing Gaussian diffusion for coordinates and categorical diffusion for discrete-valued data modalities, including chemical elements and bond types.

$$L_{t-1} = w_s(t)\Big(\lambda_x||\mathbf{X}_0 - \hat{\mathbf{X}}_0||^2 + \lambda_h \mathrm{CE}(\mathbf{H}_0, \hat{\mathbf{H}}_0) + \lambda_e \mathrm{CE}(\mathbf{E}_0, \hat{\mathbf{E}}_0)\Big), \qquad (4)$$

where CE refers to the cross-entropy loss and $(\lambda_x, \lambda_h, \lambda_e) = (3, 0.4, 2)$ are weighting coefficients adapted from Vignac et al. (2023).

In all experiments, EQGAT-diff uses 256 scalar and vector features each and 128 edge features across 12 layers of fully connected message passing. This corresponds to 12.3M trainable parameters.

We train for 200 epochs on QM9 and 400 epochs on GEOM-Drugs to achieve comparability across models while ensuring computational feasibility regarding many ablation experiments. We use fewer epochs for QM9 since the diffusion models quickly overfit such that the novelty of sampled molecules decreases. This is not the case with GEOM-Drugs.

We use the AMSGrad with a learning rate of $2 \cdot 10^{-4}$, weight-decay of $1 \cdot 10^{-12}$, and gradient clipping for values higher than ten throughout all experiments. The weights of the final model are obtained by an exponential moving average with a decay factor of 0.999.

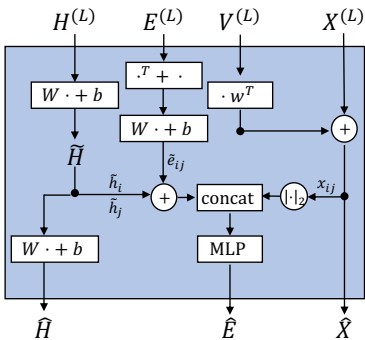

Figure 3: Prediction module that processes EQGAT-diff embeddings to obtain the predicted data modalities. The computational graph reads from top to bottom.

On the QM9 dataset, we use a batch size of 128; on the GEOM-Drugs dataset, we use an adaptive dataloader with a batch size of 800 following (Vignac et al., 2023). All models are trained on four Nvidia A100 GPUs.

For training, we use an adaptive noise schedule proposed by (Vignac et al., 2023):

$$\bar{\alpha}^t = \cos\left(\frac{\pi}{2}\frac{(t/T+s)^\nu}{1+s}\right)^2.$$

The respective scaling hyperparameter $\nu$ was set to $\nu_r = 2.5, \nu_y = 1.5, \nu_x = \nu_c = 1$ on the QM9 dataset. At the same time, for GEOM-Drugs we use $\nu_r = 2$ with $\nu_r, \nu_x, \nu_y$ and $\nu_c$ denoting atom coordinates, atom types, bond types, and charges, respectively. This noise scheduler accounts for the various variables of graph and 3D structure not being equally informative for the model and has been found by Vignac et al. (2023) to outperform the cosine schedule (Nichol & Dhariwal, 2021; Hoogeboom et al., 2022) significantly.

### A.1.2 MODEL SAMPLING

As mentioned in Sec. 5.2, the diffusion loss term $L_t = -D_{KL}[q(x_{t-1}|x_t, x_0)|p_\theta(x_{t-1}|x_t)]$ is optimized by minimizing the KL-divergence. For the case of continuous data types, i.e., coordinates, the tractable reverse distribution (Sohl-Dickstein et al., 2015; Ho et al., 2020) is

$$q(\mathbf{x}_{t-1}|\mathbf{x}_t, \mathbf{x}_0) = \mathcal{N}(\mathbf{x}_{t-1}|\mu_{t-1}(\mathbf{x}_t, \mathbf{x}_0), \Sigma_{t-1}), \tag{5}$$

with $\mu_{t-1}(\mathbf{x}_t, \mathbf{x}_0) = \frac{\sqrt{\bar{\alpha}_{t-1}}\beta_t}{1-\bar{\alpha}_t}\mathbf{x}_0 + \frac{\sqrt{\alpha_t}(1-\bar{\alpha}_{t-1})}{1-\alpha_t}\mathbf{x}_t$ and $\Sigma_{t-1} = \frac{1-\bar{\alpha}_{t-1}}{1-\bar{\alpha}_t}\beta_t\mathbf{I}$, where we assume that the coordinate matrix is vectorized to have shape $3N$.

Sampling from that reverse distribution is obtained through the denoising network that predicts the clean coordinate matrix to parameterize $p_\theta(\mathbf{x}_{t-1}|\mathbf{x}_t) = q(\mathbf{x}_{t-1}|\mathbf{x}_t, \hat{\mathbf{x}}_0)$ and sample via

$$\mathbf{x}_{t-1} = \frac{\sqrt{\bar{\alpha}_{t-1}}\beta_t}{1-\bar{\alpha}_t}\hat{\mathbf{x}}_0 + \frac{\sqrt{\alpha_t}(1-\bar{\alpha}_{t-1})}{1-\alpha_t}\mathbf{x}_t + \sqrt{\frac{1-\bar{\alpha}_{t-1}}{1-\bar{\alpha}_t}\beta_t}\cdot\epsilon_{\mathbf{CM}}, \tag{6}$$

where $\epsilon_{\mathbf{CM}} = \epsilon - \frac{1}{3N}\sum_i^{3N}\epsilon_i$ is a Gaussian noise vector with zero mean.

For discrete variables, we obtain a tractable reverse distribution that is categorical Austin et al. (2021)

$$q(\mathbf{c}_{t-1}|\mathbf{c}_0, \mathbf{c}_t) = \mathcal{C}(\mathbf{c}_{t-1}|p_{t-1}(\mathbf{c}_0, \mathbf{c}_t)), \tag{7}$$

with probability vector defined as $p_{t-1}(\mathbf{c}_0, \mathbf{c}_t) = \frac{\mathbf{c}_t\mathbf{U}_t^\top\odot\mathbf{c}_0\bar{\mathbf{U}}_{t-1}}{\mathbf{c}_0\bar{\mathbf{U}}_t\mathbf{c}_t^\top}$ where the entry $[\mathbf{U}_\mathbf{t}]_{ij}$ denotes the transition probability to jump from state $i$ to $j$ and is defined as

$$\mathbf{U}_t = (1-\beta_t)\mathbf{I}_K + \beta_t\mathbf{1}_K\tilde{\mathbf{c}}^\top = \alpha_t\mathbf{I}_K + (1-\alpha_t)\mathbf{1}_K\tilde{\mathbf{c}}^\top, \tag{8}$$

while the cumulative product after $t$ timesteps starting from 1 can be simplified to

$$\bar{\mathbf{U}}_t = \mathbf{U}_1\mathbf{U}_2\ldots\mathbf{U}_t = \bar{\alpha}_t\mathbf{I}_K + (1-\bar{\alpha}_t)\mathbf{1}_K\tilde{\mathbf{c}}^\top. \tag{9}$$

We recall that the one-hot encoding of each node or edge is perturbed independently during the forward process, such that the encoding $\mathbf{c}_t \in \{0,1\}^K$ is obtained by sampling from the categorical distribution $q(\mathbf{c}_t|\mathbf{c}_0) = \mathcal{C}(\mathbf{c}_t|\bar{\alpha}_t\mathbf{c}_0 + (1-\bar{\alpha}_t)\tilde{\mathbf{c}})$ as described in Eq. (1).

Similar to (Austin et al., 2021; Vignac et al., 2023), we obtain the reverse process for discrete data types by marginalizing the network predictions (for each node in the graph)

$$p_\theta(\mathbf{c}_{t-1}|\mathbf{c}_t) \propto \sum_{k=1}^{K} q(\mathbf{c}_{t-1}|\mathbf{c}_t, \mathbf{e}_k)\hat{c}_{0,k}, \tag{10}$$

where $\mathbf{e}_k$ is an one-hot-encoding with 1 at index $k$ and $\hat{c}_{0,k}$ is the $k$-th entry in the softmaxed probability vector $\hat{\mathbf{c}}_0$.

## A.2 METRICS

The Wasserstein distance between valencies is given as a weighted sum over the valency distributions for each atom type

$$\text{ValencyW}_1 = \sum_{x \in \text{ atom types}} p(x) \mathcal{W}_1 \left( \hat{D}_{\text{val}}(x), D_{\text{val}}(x) \right), \tag{11}$$

with $p^X(x)$ being the marginal distribution of atom types in the training set and $\hat{D}_{\text{val}}(x)$ the marginal distribution of valencies for atoms of type $x$ in the generated set and $D_{\text{val}}(x)$ the same distribution in the test set. For the bond lengths metric, a weighted sum of the distance between bond lengths for each bond type is used

$$\text{BondLenghtsW}_1 = \sum_{y \in \text{ bond types}} p(y) \mathcal{W}_1 \left( \hat{D}_{\text{dist}}(y), D_{\text{dist}}(y) \right), \tag{12}$$

where $p^Y(y)$ is the proportion of bond of types $y$ in the training set, $\hat{D}_{\text{dist}}(y)$ is the generated distribution of bond lengths for the bond of type $y$, and $D_{\text{dist}}(y)$ is the same distribution computed over the test set. Lastly, the distribution of bond angles for each atom type is a weighted sum using the proportion of each atom type in the dataset, restricted to atoms with two or more neighbors, ensuring that angles can be defined

$$\text{BondAnglesW}_1(\text{generated}, \text{target}) = \sum_{x \in \text{ atom types}} \tilde{p}(x) \mathcal{W}_1 \left( \hat{D}_{\text{angles}}(x), D_{\text{angles}}(x) \right), \tag{13}$$

with $\tilde{p}^X(x)$ denoting the proportion of atoms of types $x$ in the training set, and $D_{\text{angles}}(x)$ the distribution of geometric angles of the form $\angle (\boldsymbol{r}_k - \boldsymbol{r}_i, \boldsymbol{r}_j - \boldsymbol{r}_i)$, where $i$ is an atom of type $x$, and $k$ and $j$ are neighbors of $i$ (Vignac et al., 2023).

## A.3 RESULTS AND DETAILS

We visualize the empirical distribution of the number of atoms and the chemical composition for the QM9, GEOM-Drugs, and PubChem3D datasets in Figure 4. For PubChem3D, we show the empirical distribution for the datasets with implicit and explicit hydrogens.

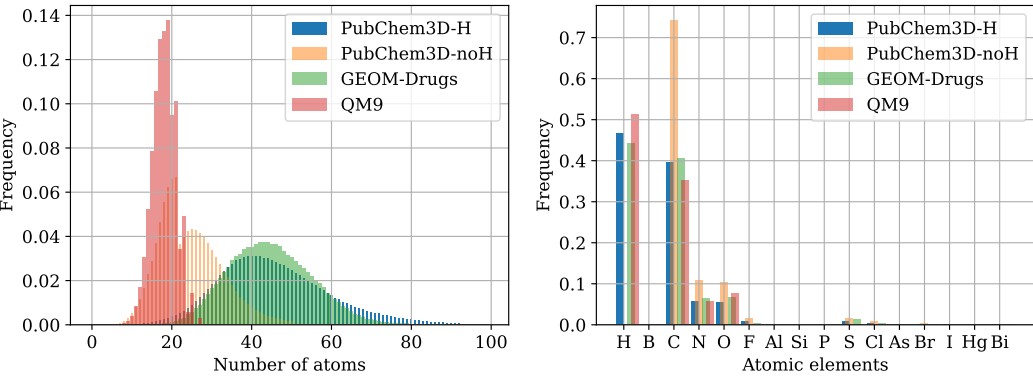

Figure 4: Empirical distributions over QM9, GEOM-Drugs, and PubChem3D with implicit and explicit hydrogens. a) Frequency for the number of atoms. b) Frequency for atomic elements.

## A.4 TIME-DEPENDENT LOSS WEIGHTING

### A.4.1 LOSS WEIGHTING AND FINE-TUNING

In the study in Section A.4, we conducted an ablation analysis to evaluate the efficacy of loss weighting, comparing two weighting strategies denoted as $w_s(t)$ and $w_u$, across different subsets (25, 50, 75, and 100%) of the QM9 and GEOM-Drugs datasets. In Fig. 5 the truncated loss weighting is

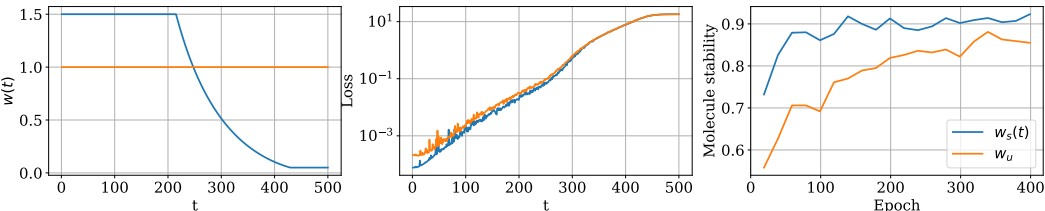

Figure 5: Comparison of EQGAT-diff trained with $w_s(t)$ and $w_u$, respectively, on GEOM-Drugs. a) Uniform ($w_u$) versus modified SNR(t) loss-weighting ($w_s(t)$). b) Unweighted prediction errors for models trained with $w_u$ or $w_s(t)$ loss-weightings over increasing timesteps. c) Comparison between $w_u$ and $w_s(t)$ regarding molecule stability convergence during training.

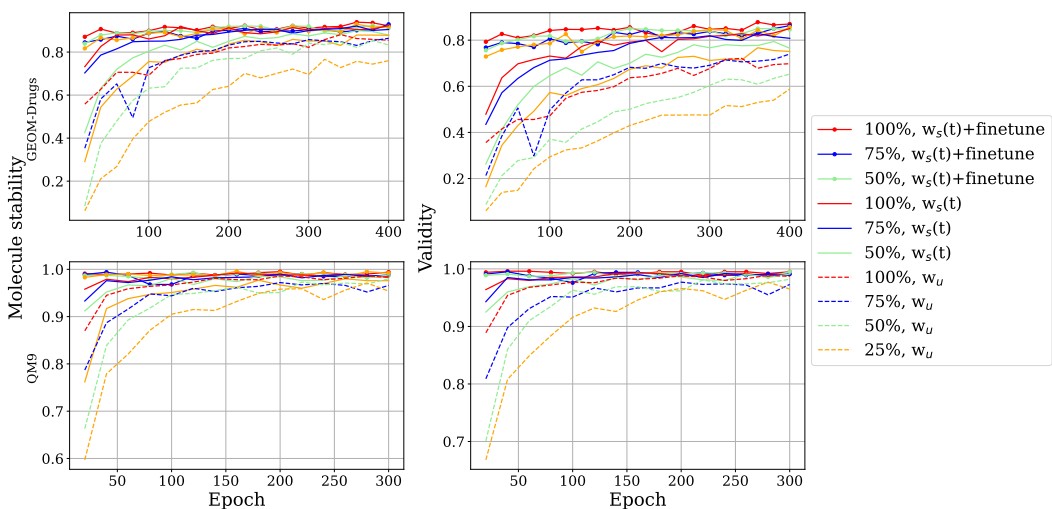

Figure 6: Comparison of different models and data subsets for training on GEOM-Drugs and QM9, respectively. The dotted, solid lines depict the fine-tuned model using $w_s(t)$-weighting. The solid lines show the model using $w_s(t)$-weighting and the dashed lines show the model trained without loss-weighting. While training, after every 20 epochs, 1000 sampled molecules are evaluated on molecule stability and validity.

depicted (left) besides the effect on the loss for lower timesteps illustrating the unweighted loss over time steps for a batch of 128 molecules, where the model trained with $w_s(t)$ achieves lower prediction error for steps closer to 1 (middle) and the effect on the molecule stability while training showing better performance and faster training convergence for molecule stability when using $w_s(t)$ (right).

As illustrated in Figure 6, applying loss weighting using $w_s(t)$ consistently results in performance enhancements for the model. These enhancements are characterized by accelerated training convergence, leading to improved molecule stability and validity, even when the model operates on smaller subsets of the data. Notably, in the case of GEOM-Drugs, when trained with only 25% of the data and optimized with $w_s(t)$ (indicated by the yellow solid line), the model exhibits convergence behavior similar to that of the model trained on 100% of the data with uniform weighting $w_u$ (indicated by the yellow dashed line). Furthermore, fine-tuning leads to superior performance (dotted, solid lines). After just 20 epochs of fine-tuning and only using 25% of the data, the model already outperforms all its counterparts on molecule stability and validity even when they are trained on 100% of the data and holds for both the GEOM-Drugs (first row) and QM9 (second row) datasets. Our findings, as summarized in Table 6, underscore the critical role of loss weighting using $w_s(t)$ in the training of diffusion models for molecular data and also highlight the importance of pre-training, especially when the target distributions are small and do not contain many data points.

Table 6: Comparison of EQGAT-diff on QM9 and GEOM-Drugs trained on subsets of 25, 50 and 75% of the data. We report the mean values over five runs of Molecular Stability (Mol. Stability), Validity, and the number of Connected Components (Connect. Comp.) for training from scratch with and without modified SNR(t) weighting and compare it with the performance of the fine-tuned model (SNR(t)+fine-tune). The best results are written in bold, and results with overlapping margins of errors are underlined. The margin of error for the 95% confidence level is given as subscripts.

| | | QM9 | | | GEOM-Drugs | | |
|---|---|---|---|---|---|---|---|
| | Subset | Mol. Stability | Validity | Connect. Comp. | Mol. Stability | Validity | Connect. Comp. |
| $w_u$ | 25% | $96.01_{\pm0.22}$ | $96.68_{\pm0.24}$ | $99.59_{\pm0.05}$ | $74.12_{\pm0.29}$ | $51.32_{\pm0.38}$ | $68.88_{\pm0.25}$ |
| | 50% | $96.84_{\pm0.16}$ | $97.45_{\pm0.15}$ | $99.75_{\pm0.03}$ | $85.20_{\pm0.27}$ | $64.19_{\pm0.39}$ | $82.76_{\pm0.26}$ |
| | 75% | $96.19_{\pm0.18}$ | $96.83_{\pm0.17}$ | $99.84_{\pm0.03}$ | $87.08_{\pm0.33}$ | $74.27_{\pm0.29}$ | $88.69_{\pm0.29}$ |
| | 100% | $97.39_{\pm0.23}$ | $97.99_{\pm0.20}$ | $99.70_{\pm0.03}$ | $87.59_{\pm0.19}$ | $71.44_{\pm0.22}$ | $86.57_{\pm0.33}$ |
| $w_s(t)$ | 25% | $97.34_{\pm0.15}$ | $97.77_{\pm0.09}$ | $99.81_{\pm0.03}$ | $88.39_{\pm0.39}$ | $75.44_{\pm0.46}$ | $85.35_{\pm0.51}$ |
| | 50% | $98.32_{\pm0.11}$ | $98.65_{\pm0.07}$ | $99.93_{\pm0.03}$ | $89.41_{\pm0.26}$ | $77.21_{\pm0.28}$ | $89.43_{\pm0.23}$ |
| | 75% | $98.45_{\pm0.08}$ | $98.77_{\pm0.04}$ | $99.93_{\pm0.02}$ | $91.88_{\pm0.20}$ | $82.77_{\pm0.16}$ | $93.39_{\pm0.20}$ |
| | 100% | $98.68_{\pm0.11}$ | $98.96_{\pm0.07}$ | $99.94_{\pm03}$ | $91.66_{\pm0.14}$ | $84.02_{\pm0.19}$ | $95.08_{\pm0.12}$ |
| $w_s(t)$+ fine-tune | 25% | $99.00_{\pm0.13}$ | $99.24_{\pm0.10}$ | $99.96_{\pm0.01}$ | $90.82_{\pm0.67}$ | $83.01_{\pm1.30}$ | $93.77_{\pm0.76}$ |
| | 50% | $\mathbf{99.21}_{\pm0.09}$ | $\mathbf{99.41}_{\pm0.07}$ | $\mathbf{99.96}_{\pm0.01}$ | $91.24_{\pm0.82}$ | $83.83_{\pm1.51}$ | $94.66_{\pm0.77}$ |
| | 75% | $98.79_{\pm0.10}$ | $99.12_{\pm0.12}$ | $99.95_{\pm0.03}$ | $\mathbf{92.97}_{\pm0.15}$ | $\mathbf{86.51}_{\pm0.17}$ | $\mathbf{95.92}_{\pm0.14}$ |
| | 100% | $98.94_{\pm0.07}$ | $99.28_{\pm0.09}$ | $99.95_{\pm0.02}$ | $\mathbf{93.19}_{\pm0.07}$ | $\mathbf{86.83}_{\pm0.20}$ | $\mathbf{96.31}_{\pm0.21}$ |

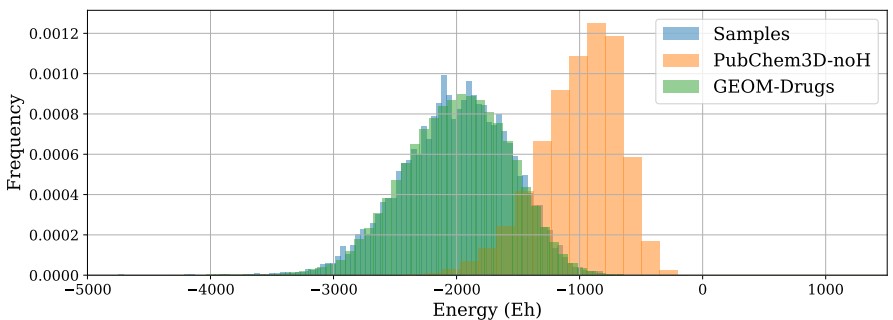

Figure 7: Comparison of the energy distributions calculated using xTB-GFN2 Bannwarth et al. (2019) for the GEOM-Drugs training dataset against the energies of sampled molecules. We also provide the energy distribution of PubChem3D (implicit hydrogens) to showcase the distribution shift; for quantum physics-based software, those molecules appear to be radicals, and hence, the energy distribution is shifted towards high energies. Nevertheless, the model effectively has to do this shift while fine-tuning.

### A.5 PRE-TRAINING ON PUBCHEM3D

To emphasize more the capability of the model to learn the underlying data distribution, we follow (Hoogeboom et al., 2022) and plot the distribution of energies for sampled molecules of a model trained on GEOM-Drugs against the energy distribution of the training dataset, as shown in Fig. 7. We observe that EQGAT-diff learns the training distribution well, showing a high overlap. Furthermore, to highlight the shift in physical space the diffusion model has to perform while fine-tuning, we also report the energy distribution of PubChem3D with implicit hydrogens. All energies were calculated using the semi-empirical xTB-GFN2 software (Bannwarth et al., 2019).

We also pre-trained a model on the PubChem3D dataset with explicit hydrogens. Interestingly, as shown in Tab. 7 we see a decrease in performance for the model that is fine-tuned on the pre-training with explicit hydrogens compared to the model using implicit hydrogens, even though pre-training with explicit hydrogens takes almost three times as long. We suspect that when using explicit hydrogens in pre-training, the model overfits too much on the PubChem3D data distribution, having a more challenging time transferring to the GEOM-Drugs distribution.

We subsampled 1M molecules from PubChem3D and GEOM-Drugs and enumerated over bonds of selected pair atoms including carbon, hydrogen, nitrogen and oxygen atoms. We computed distances

Table 7: Comparison of EQGAT-diff pre-trained with or without explicit hydrogens on PubChem3D and fine-tuned on GEOM-Drugs for 400 epochs. We report the mean values over five runs of selected evaluation metrics with the margin of error for the 95% confidence level given as subscripts. The best results are in bold.

| Pretraining | Mol. Stab. ↑ | Validity ↑ | Connect. Comp. ↑ |
|---|---|---|---|
| PubChem3D-noH | $\mathbf{93.19}_{\pm 0.07}$ | $\mathbf{86.83}_{\pm 0.20}$ | $\mathbf{96.31}_{\pm 0.21}$ |
| PubChem3D-H | $92.70_{\pm 0.09}$ | $85.46_{\pm 0.19}$ | $94.78_{\pm 0.19}$ |

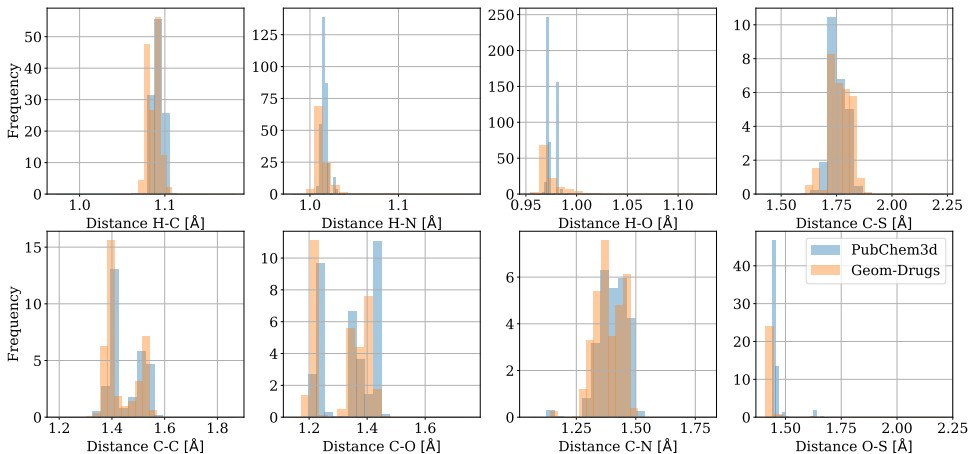

Figure 8: Selected atom-pair distance distribution on PubChem3D and GEOM-Drugs.

and noticed that the hydrogen-oxygen distance distribution in PubChem3D seems to have a smaller variance than GEOM-Drugs in the last panel of Figure 8.

## A.6 EQGAT-DIFF VS MIDI

We found EQGAT-diff outperforming MiDi by a large margin across both datasets, QM9 and GEOM-Drugs, and all metrics. In Fig. 9, we underpin this observation by comparing training curves of $\text{EQGAT}_{\text{disc}}^{x0}$ and the MiDi model, observing that our model not only outperforms MiDi on molecule stability, validity and in the adaptation to the underlying data distribution, but also converges significantly faster. $\text{EQGAT}_{\text{disc}}^{x0}$ converges to SOTA performance already after 150-200 epochs, while MiDi needs roughly 700 epochs weakly indicating convergence but to lower values.

Furthermore, EQGAT-diff needs ∼5 minutes per epoch using four Nvidia A100 GPUs, adaptive dataloading (taken from the MiDi code based on `pyg.loader.Collater`) with a batch size of 200 per GPU. In contrast, MiDi takes ∼12 minutes, so EQGAT-diff is more than twice as fast.

For training MiDi, we used the official codebase on GitHub [1] and the given hyperparameter settings but trained on four Nvidia A100 GPUs (instead of two). As seen in Tab.3 and shown here in Fig. 9b, we could not reproduce the results reported in the paper. We also re-evaluated the checkpoint given on GitHub and again could not confirm the reported results.

## A.7 EQGAT-DIFF WITH IMPLICIT HYDROGENS ON GEOM-DRUGS

We trained EQGAT-diff on GEOM-Drugs with implicit hydrogens. To this end, we pre-process the GEOM-Drugs dataset using the RDKit and remove hydrogens from a molecule object `mol` using the `Chem.RemoveHs` function, with subsequent kekulization `Chem.Kekulize`. We list the evaluation results of models $\text{EQGAT}_{disc}^{x_0}$ and $\text{EQGAT}_{cont}^{x_0}$ in Table 8 below. We discover that the Gaussian and categorical diffusion for the $x_0$ parameterization achieves similar performance

---

[1]https://github.com/cvignac/MiDi

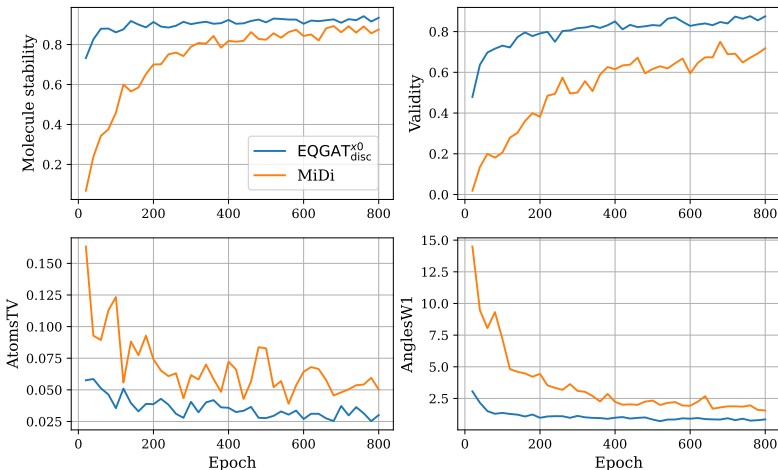

Figure 9: Comparison between EQGAT-diff and MiDi for training on GEOM-Drugs. We compare both models regarding a) molecular stability, b) validity, c) AtomsTV, and d) AnglesW1 while training by sampling 1000 molecules every 20 epochs over 800 epochs of training. For molecular stability and validity, higher is better; for AtomsTV and AnglesW1, lower is better.

related to validity and connected components. At the same time, the Wasserstein-1 distance on the histograms for empirical bond angles is lower for the generated set from $\text{EQGAT}_{disc}^{x_0}$ to the histogram of the reference set.

Table 8: Comparison of EQGAT-diff with implicit hydrogens on GEOM-Drugs for 400 epochs. We report the mean values over five runs of selected evaluation metrics with the margin of error for the 95% confidence level given as subscripts. The best results are in bold.

| Model | Validity ↑ | Connect. Comp. ↑ | BondLengths W1 ↓ | BondAngles W1 ↓ |
|---|---|---|---|---|
| $\text{EQGAT}_{disc}^{x_0}$ | $\mathbf{98.29}_{\pm 0.08}$ | $\mathbf{98.90}_{\pm 0.10}$ | $\mathbf{0.59}_{\pm 0.62}$ | $\mathbf{0.44}_{\pm 0.01}$ |
| $\text{EQGAT}_{cont}^{x_0}$ | $\mathbf{98.48}_{\pm 0.14}$ | $98.36_{\pm 0.09}$ | $1.34_{\pm 0.07}$ | $0.56_{\pm 0.03}$ |

Table 9: Classifier-guidance on EQGAT-diff to shift the reverse sampling towards low or high polarizability values. We report the mean polarizability values of sampled molecules with standard deviations as subscripts.

| Guidance | Polarizability |
|---|---|
| Minimization | $195.19_{\pm 4.9}$ |
| Maximization | $400.21_{\pm 8.3}$ |

## A.8 CLASSIFIER GUIDANCE FOR CONDITIONAL MOLECULE DESIGN

For conditional molecule design, we can use a trained unconditional EQGAT-diff model together with classifier-guidance, as proposed in (Dhariwal & Nichol, 2021), to steer the generation of samples using the gradient of an external classifier/regressor during the reverse sampling trajectory from noise to data. As a proof of concept, we explored classifier-guidance to generate molecules optimizing for low/high polarizability showing promising results. For this, we trained a polarizability regressor model and applied it to the reverse sampling of an unconditional EQGAT-diff model testing for guiding towards low and high polarizability values, respectively. Afterwards we re-calculate the polarizability of all sampled molecules for both cases and compared the mean values. In Tab.

9 we summarize the results. The mean value of the GEOM-Drugs training dataset is $245.9 \pm 41.9$. Hence, we see that we can successfully push the distribution of sampled molecules in the respective direction.

## A.9 COMPARISON TO MOLDIFF

We compare against MolDiff Peng et al. (2023) by utilizing their evaluation pipeline that includes post-processing steps on the generated molecules to potentially fix valency and aromaticity issues when parsing into the RDKit. Selecting the $5 \times 10,000$ generated samples from our best performing model $\text{EQGAT}^{x0,af,ft}_{\text{disc}}$ we report mean validity and mean success rate in Table 10. As shown, our proposed best-performing EQGAT-diff model achieves superior performance over MolDiff in generating chemically valid molecules but has lower diversity, which we believe is caused by longer training time on our side. However, we believe the generative model should be able to faithfully sample molecules that satisfy valency constraints, as it was also trained in such data. Suppose we do not employ the post-processing scheme from MolDiff and determine the validity by parsing the generated molecule into RDKit's sanitization pipeline. In that case, EQGAT-diff obtains a mean validity of 0.916 and a mean success rate of 0.887. This shows that the post-processing applied in MolDiff substantially impacts model evaluation.

Table 10: Evaluation metrics from EQGAT-diff against MolDiff.

| Model | EQGAT-diff | MolDiff |
|---|---|---|
| Validity | **0.998** | 0.947 |
| Connectivity | **0.968** | 0.908 |
| Succ. Rate | **0.966** | 0.860 |
| Novelty | 1.000 | 1.000 |
| Uniqueness | 1.000 | 1.000 |
| Diversity | 0.320 | **0.422** |

## A.10 IMPROVED SAMPLING TIME

We experimented with the DDIM Song et al. (2021a) sampling algorithm known for enhancing inference/sampling time in diffusion models trained via the standard DDPM procedure in image processing. The difference between DDIM and DDPM lies in the sampling algorithm, which we believe could also be applied in our molecular data setting. However, our best-performing scenario utilizes the $x_0$ parameterization to preserve the correct data modalities for coordinates, atom, and bond features. Hence, applying DDIM directly to discrete-valued data modalities is not straightforward. We restricted DDIM to continuous coordinate updates, while discrete-valued data modalities follow the approach outlined by (Austin et al., 2021) and explained in our Appendix in Eq. (10). Table 11 compares the evaluation performance of our base models when generating samples using DDIM or DDPM for varying numbers of reverse sampling steps $500, 250, 167$. Given that all models underwent training with $T = 500$ discretized timesteps, we conducted DDIM sampling every 2 or 3 steps of the reversed trajectory starting from index 500. Notably, we observed that employing DDIM did not enhance the quality of molecule generation with fewer sampling steps (250 or 166) compared to the 500 steps the models were trained on.

Another way to enhance sampling time, is to train the diffusion models with less discretized timesteps. We performed additional experiments and trained $\text{EQGAT}^{x_0}_{disc}$ with $T = 100$ timesteps using the uniform and truncated SNR(t) loss weighting. The rationale behind these experiments is to assess how the reduced number of timesteps affects performance while enabling faster inference time. We compare against the two corresponding models trained with $T = 500$ timesteps and observe that the model trained with truncated SNR(t) loss weighting over $T = 100$ timesteps performs better than the model trained with $T = 500$ timesteps but uniform loss weighting as illustrated in Figure 10. This result clearly speaks for the usage of the proposed loss weighting while also achieving a diffusion model that has a faster sampling time using 100 reverse sampling steps only, i.e. around 5x faster sampling time. Comparing the two models trained with truncated SNR(t) loss weighting, we notice that the model trained with $T = 500$ discretized steps still performs better than the identical model but trained with $T = 100$ timesteps.

Table 11: Sampling results of trained models for DDPM and DDIM for 500 Molecules.

| Model | Steps | Sampling | Runtime | Mol. Stability | Validity | AnglesW1 |
|---|---|---|---|---|---|---|
| Discrete-SNR(t) | 500 | DDPM | 26min | 0.9160 | 0.8100 | 0.83 |
| Continuous-SNR(t) | 500 | DDPM | 27min | 0.8920 | 0.7600 | 0.90 |
| Discrete-Uniform | 500 | DDPM | 26min | 0.8600 | 0.5960 | 1.36 |
| Discrete-SNR(t) | 250 | DDIM | 13min | 0.6580 | 0.6260 | 2.26 |
| Continuous-SNR(t) | 250 | DDIM | 13min | 0.5680 | 0.3920 | 3.95 |
| Discrete-Uniform | 250 | DDIM | 13min | 0.5400 | 0.3880 | 3.21 |
| Continuous-SNR(t) | 250 | DDPM | 13min | 0.5160 | 0.3400 | 4.74 |
| Discrete-SNR(t) | 250 | DDPM | 13min | 0.4860 | 0.4600 | 4.60 |
| Discrete-Uniform | 250 | DDPM | 14min | 0.2620 | 0.1940 | 7.60 |
| Discrete-SNR(t) | 166 | DDIM | 9min | 0.1980 | 0.2280 | 5.58 |
| Discrete-Uniform | 166 | DDIM | 9min | 0.1240 | 0.1080 | 8.24 |
| Continuous-SNR(t) | 166 | DDPM | 8min | 0.1000 | 0.0380 | 13.68 |
| Continuous-SNR(t) | 166 | DDIM | 9min | 0.0900 | 0.0520 | 14.11 |
| Discrete-SNR(t) | 166 | DDPM | 9min | 0.0660 | 0.5200 | 12.21 |
| Discrete-Uniform | 166 | DDPM | 9min | 0.0200 | 0.0120 | 14.83 |

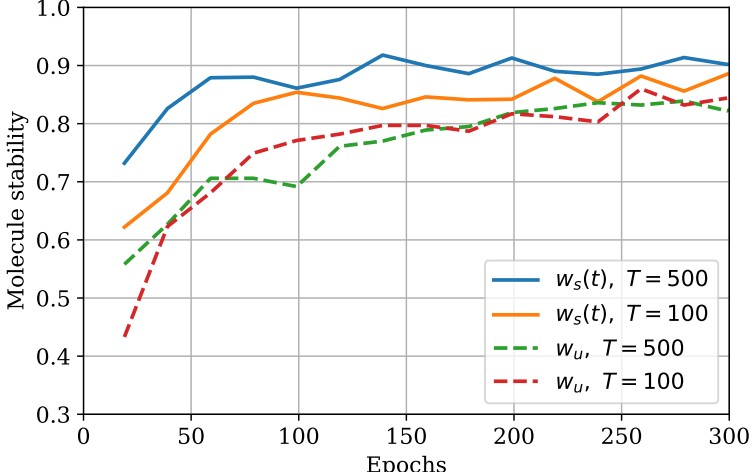

Figure 10: Molecule stability learning curves for diffusion models trained with $T = 500$ and $T = 100$ discrete timesteps. Again, we observe that the truncated SNR(t) loss weighting $w_s(t)$ greatly improves performance.

