# OpenReview forum: "Navigating the Design Space of Equivariant Diffusion-Based Generative Models for De Novo 3D Molecule Generation"
_ICLR.cc/2024/Conference — ICLR 2024 poster_

### Official Review · Reviewer_WVdk · 2023-10-19

**Soundness:** 2 fair
**Presentation:** 1 poor
**Contribution:** 1 poor
**Rating:** 3
**Confidence:** 4

**Summary:**

The manuscript introduces the EQGAT-diff model, an adaptation of the previously proposed EQGAT network, to model denoising diffusion generation. While there isn't a significant technical innovation, the model does exhibit enhanced empirical performance based on the provided metrics. The primary focus of the model is the unconditional semi de novo generation of 3D molecular structures. Notably, the relevance of generating valence-correct molecular structures without specific conditions is not clear. The authors highlight potential applications in structure-based drug design, yet the model's current format doesn't seem optimized for this purpose. Additionally, there is no mention of code availability. The presentation and writing of the article require improvement. In its current state, the manuscript lacks a meaningful technical contribution and seems to address an ill-defined task. I suggest that the authors refine their model for structure-based drug design and consider resubmitting for a future conference.

**Strengths:**

## Empirical Performance
The model demonstrates superior performance based on the evaluation metrics presented.

**Weaknesses:**

## Task Relevance
The de novo generation of 3D molecular structures, as currently presented, appears to lack inherent value.

## Lack of Technical Novelty
The work does not introduce new technical advancements.

## Code Unavailability
Absence of open-source code hinders replication and comparison efforts.

## Writing Quality
The referencing is inconsistent, with many citations missing journal or conference details. There are incorrect uses of quotation marks, and figure clarity is compromised, diminishing the manuscript's overall professionalism.

**Questions:**

It would be beneficial for the authors to compare their 3D molecule generation model with a 2D molecule generation, specifically using OpenEye’s OMEGA, which is cited as the ground truth for conformation generation. Would the EQGAT-diff model maintain its efficiency advantage in such a comparison?

---

> ### Author Response · Authors · 2023-11-17
> **Answer to reviewer WVdk**
>
> ### Weaknesses
>
> ####  Task Relevance
>
> > The de novo generation of 3D molecular structures, as currently presented, appears to lack inherent value.
>
> ### Answer
> Our study extensively examines various model design choices for 3D molecular data. Initially, we focused on unconditional 3D molecule generation to gain relevant insights into essential design decisions, such as tailoring fast and powerful equivariant deep learning models, incorporating categorical diffusion on discrete data modalities, and a task-specific loss weighting function. Our findings show new state-of-the-art results by a large margin while being significantly faster in training and inference (Appendix Fig. 9 and Fig. 10). Thus, this work helps build the foundation for further investigations in generative chemistry.
>
> A critical aspect of it is, e.g., structure-based conditional molecule generation. Here, sampling molecules with specific quantum- and physicochemical properties in the presence of known protein receptors necessitates accurate, reliable, and fast 3D molecular modeling.
>
> For conditional molecule design, we can use a trained unconditional EQGAT-diff model together with classifier-guidance, as proposed in [1], to steer the generation of samples using the gradient of an external classifier/regressor during the reverse sampling trajectory from noise to data. As a proof of concept, we explored classifier-guidance to generate molecules optimizing for low/high polarizability showing promising results.
>
> To further highlight the importance of our findings, we provide results on structure-based de novo ligand generation. Here, we evaluated EQGAT-diff on the Crossdocked dataset [2] and found that SNR-t weighting and finetuning are crucial components confirming our previous results.
>
> #### Lack of Technical Novelty
>
> >The work does not introduce new technical advancements.
>
> ### Answer
> In this work, we provide a profound contribution by rigorous experimental design containing many important ablations.
> We think that in science, experimental validation and exploration are at least as important as technical novelty. Especially in ML, we find that proper evaluation by in-depth experimentation and ablation is often harmed when focusing on delivering theoretical or architectural novelty. This work's main idea and driver was the belief that current state-of-the-art 3D diffusion models were not sufficiently examined, resulting in significant flaws, e.g., in the design of larger molecular structures. Thus, we focus on a concise experimental investigation of well-known theoretical findings. We show that by exploring the diffusion design space, it is possible to vastly increase the model's generative performance - e.g., compared to the latest SOTA[3], our model increases the validity on sampled molecules on average by roughly 25\%; we increase the molecular stability (3D metric) by roughly 240\% compared to EDM [4].
>
>
> #### Code Unavailability
> >Absence of open-source code hinders replication and comparison efforts.
>
> ### Answer
> The code will be made public as soon as our internal clearance is finished in the upcoming days.
> We have developed a code base that is easy to use and allows full reproducibility of the underlying work.
>
> #### Writing Quality
> >The referencing is inconsistent, with many citations missing journal or conference details. There are incorrect uses of quotation marks, and figure clarity is compromised, diminishing the manuscript's overall professionalism.
>
> ### Answer
> We checked the references carefully, added missing journal and conference details, and addressed the concerns raised about incorrect uses of quotation marks. We updated the figures to improve the overall readability of the manuscript.

---

> ### Author Response · Authors · 2023-11-17
> **Continued**
>
> ### Questions
> >It would be beneficial for the authors to compare their 3D molecule generation model with a 2D molecule generation, specifically using OpenEye’s OMEGA, which is cited as the ground truth for conformation generation. Would the EQGAT-diff model maintain its efficiency advantage in such a comparison?
>
> ### Answer
> EQGAT-diff is designed and trained to perform de novo molecule generation. Our model is not conditioned on a molecular topology and composition and, thus, is not intended to sample conformers.
> Despite this, to our knowledge, there is no SMILES strings or graph-based generative model to which we could compare our results on GEOM-Drugs.
> The task conditional sampling of molecules, where the condition is inherently connected to the 3D structure, e.g., a quantum property or a binding affinity of a ligand to a receptor, would not be feasible with the 2D graph and conformer sampling approach. With our work, we want to lay a solid foundation of selected design components to tackle molecule generation in  3D space, with the further goal to apply such in the structured-based conditional setting.
>
> In light of the mentioned improvements, we ask the reviewer to consider raising the score.
>
> #### References
>
> 1] Prafulla Dhariwal and Alexander Quinn Nichol. Diffusion models beat GANs on image synthesis. In A. Beygelzimer, Y. Dauphin, P. Liang, and J. Wortman Vaughan (eds.), Advances in Neural Information Processing Systems, 2021. URL https://openreview.net/forum?id=AAWuCvzaVt.
>
> [2] Three-Dimensional Convolutional Neural Networks and a Cross-Docked Data Set for Structure-Based Drug Design Paul G. Francoeur, Tomohide Masuda, Jocelyn Sunseri, Andrew Jia, Richard B. Iovanisci, Ian Snyder, and David R. Koes Journal of Chemical Information and Modeling 2020 60 (9), 4200-4215DOI: 10.1021/acs.jcim.0c00411
>
> [3] Clement Vignac, Nagham Osman, Laura Toni, and Pascal Frossard. Midi: Mixed graph and 3d denoising diffusion for molecule generation. In Danai Koutra, Claudia Plant, Manuel Gomez Rodriguez, Elena Baralis, and Francesco Bonchi (eds.), Machine Learning and Knowledge Discovery in Databases: Research Track - European Conference, ECML PKDD 2023, Turin, Italy, September 18-22, 2023, Proceedings, Part II, volume 14170 of Lecture Notes in Com- puter Science, pp. 560–576. Springer, 2023. doi: 10.1007/978-3-031-43415-0\ 33. URL https://doi.org/10.1007/978-3-031-43415-0_33
>
> [4] Emiel Hoogeboom, V ́ıctor Garcia Satorras, Cl ́ement Vignac, and Max Welling. Equivariant diffu- sion for molecule generation in 3D. In Kamalika Chaudhuri, Stefanie Jegelka, Le Song, Csaba Szepesvari, Gang Niu, and Sivan Sabato (eds.), Proceedings of the 39th International Confer- ence on Machine Learning, volume 162 of Proceedings of Machine Learning Research, pp. 8867–8887. PMLR, 17–23 Jul 2022. URL https://proceedings.mlr.press/v162/hoogeboom22a.html

---

> > ### Comment · Reviewer_WVdk · 2023-11-21
> > **Response to authors**
> >
> > Thank you to the authors for their response. Regarding the significance of the task, I believe the authors also acknowledge that the mere generation of 3D molecules may potentially aid in drug discovery, but in itself is not inherently meaningful. This point must be made clear. Additionally, although numerous articles have explored 3D molecular generation, to date, they have not provided any practical aid, which makes the feasibility of this approach questionable. As mentioned by the authors, working in conjunction with classifier guidance is feasible but evidently inefficient. Therefore, I suggest that the authors should at least compare their method with a 2D-based goal-oriented generation approach, then generate conformers using OpenEye’s OMEGA post-generation (a very simple and commonly used baseline). This would offer a basic assessment of potential utility. Consequently, while the addition of Section 8 is beneficial, it is insufficient.
> >
> > On another note, Section 8 merely compares the proposed model using metrics like molecular validity, without measuring truly bioactivity-related indicators such as affinity, and no other structure-based drug design methods are compared. I suggest focusing more on the structure-condition 3D generation but implementing such changes would lead to a complete revision of the manuscript. Therefore, I believe that in its current form, this article is not suitable for publication in ICLR.

---

> > > ### Author Response · Authors · 2023-11-21
> > > **Answer to reviewer WVdk**
> > >
> > > We thank the reviewer for the feedback.
> > >
> > > *Regarding the significance of the task, I believe the authors also acknowledge that the mere generation of 3D molecules may potentially aid in drug discovery, but in itself is not inherently meaningful*
> > >
> > > ## Answer
> > > Unfortunately, we do not understand the meaning of this comment. If the task potentially aids the drug discovery process, it is also potentially meaningful.
> > >
> > > *Additionally, although numerous articles have explored 3D molecular generation, to date, they have not provided any practical aid, which makes the feasibility of this approach questionable.*
> > >
> > > ## Answer
> > > It might be true that 3D (small) molecule generation tools have not provided any practical aid (yet). Nevertheless, works like DiffDock [1] have proven the effectiveness of directly modelling 3D molecules and the recent updates to Alphafold [2] and the publication of RoseTTAFold All-Atom [3] have shown the growing relevance and importance of incorporating small molecules.
> > >
> > > But, the potential lack of applications of (de novo) 3D molecule generation tools might stem from the fact that current generative models have shown unsatisfactory performance in learning more complex data distributions and in reliably generating valid molecules, even for the unconditional molecule generation (EDM).
> > >
> > > Thus, further research in this domain starting with unconditional molecule generation is necessary, and the intention of our work is to push the boundaries towards higher accuracy and hence higher applicability. By thorough experimental design, as outlined already, we built a model and framework, that provides a state-of-the-art blueprint to build generative diffusion models for unconditional 3D molecule generation significantly improving recent publications in many relevant metrics. By this, our work enables an important step towards adaption of 3D generative models.
> > > During revision, we also added 3D conditional molecular generation results to validate our framework. This shows that we can seamlessly apply our findings also to structure-based tasks that are of high relevance in practice.
> > >
> > > *As mentioned by the authors, working in conjunction with classifier guidance is feasible but evidently inefficient.*
> > >
> > > ## Answer
> > > We disagree with the reviewer that classifier-guidance is "feasible, but evidently inefficient". Recent work like GaUDI [4] shows an efficient application of classifier-guidance for material discovery. In our experiments, besides classifier-free guidance (which is also easily doable in our framework), classifier-guidance provides an elegant and simple way to apply conditions on unconditionally trained models. This also allows for multi-property optimization. Hence, the only bottleneck is the availability of large and diverse datasets providing meaningful (experimental) labels/measurements, like for training ADMET regression/classifier models. If available, this would provide a potentially powerful setting.

---

> ### Author Response · Authors · 2023-11-21
> **Continued**
>
> *Therefore, I suggest that the authors should at least compare their method with a 2D-based goal-oriented generation approach, then generate conformers using OpenEye’s OMEGA post-generation (a very simple and commonly used baseline). This would offer a basic assessment of potential utility. Consequently, while the addition of Section 8 is beneficial, it is insufficient.*
>
> ## Answer
> We do not know about any 2D(graph?)-based "goal-oriented" generation approach. DiGress [5] is a diffusion model based on graph input, but this comes with the problem that graphs have limited representational power [6], which must be toned down, e.g., with expensive spectral features via eigendecomposition. E(3) Equivariant models working on point clouds are inherently more expressive and additional post-generation via OMEGA is not necessary (despite the fact that OMEGA does not provide quantum-accurate conformers, but the generative 3D model can be trained on such data, hence being more accurate than OMEGA - and as we show in our work, pre-training on large databases built with a low level of theory, here OMEGA, is sufficient to fine-tune on data with a higher level of theory). Working on graph-based models with post-generation does also not allow for direct 3D conditional sampling like proven to work well in structure-based tasks [8, 9, 10].
>
> *On another note, Section 8 merely compares the proposed model using metrics like molecular validity, without measuring truly bioactivity-related indicators such as affinity, and no other structure-based drug design methods are compared. I suggest focusing more on the structure-condition 3D generation but implementing such changes would lead to a complete revision of the manuscript. Therefore, I believe that in its current form, this article is not suitable for publication in ICLR.*
>
> ## Answer
> In the updated manuscript, we report the QVina docking score as an indicator of binding affinity, showing that our model outperforms competing models on this task. This serves as an additional proof for the relevance of the reported findings strengthening the proposed framework.
>
> &nbsp;
> &nbsp;
>
> In the light of the above mentioned improvements, we ask the reviewer to consider raising the score.
>
> &nbsp;
> &nbsp;
>
> ## References
>
> [1] DiffDock: https://openreview.net/forum?id=kKF8_K-mBbS
>
> [2] Alphafold-update: https://deepmind.google/discover/blog/a-glimpse-of-the-next-generation-of-alphafold/
>
> [3] Rohith Krishna, Jue Wang, Woody Ahern, Pascal Sturmfels, Preetham Venkatesh, Indrek Kalvet, Gyu Rie Lee, Felix S Morey-Burrows, Ivan Anishchenko, Ian R Humphreys, Ryan McHugh, Dionne Vafeados, Xinting Li, George A Sutherland, Andrew Hitchcock, C Neil Hunter, Minkyung Baek, Frank DiMaio, David Baker
> bioRxiv 2023.10.09.561603; doi: https://doi.org/10.1101/2023.10.09.561603
>
> [4] Weiss, T., Mayo Yanes, E., Chakraborty, S. et al. Guided diffusion for inverse molecular design. Nat Comput Sci 3, 873–882 (2023). https://doi.org/10.1038/s43588-023-00532-0
>
> [5] DiGress: https://openreview.net/forum?id=UaAD-Nu86WX
>
> [6] Keyulu Xu, Weihua Hu, Jure Leskovec, and Stefanie Jegelka. How powerful are graph neural networks? In International Conference on Learning Representations, 2019. URL https: //openreview.net/forum?id=ryGs6iA5Km. 2, 5
>
> [7] Dominique Beaini, Saro Passaro, Vincent Le ́tourneau, Will Hamilton, Gabriele Corso, and Pietro Lio`. Directional graph networks. In International Conference on Machine Learning, pp. 748– 758. PMLR, 2021. 6
>
> [8] Xingang Peng, Shitong Luo, Jiaqi Guan, Qi Xie, Jian Peng, and Jianzhu Ma. Pocket2Mol: Efficient molecular sampling based on 3D protein pockets. In Kamalika Chaudhuri, Stefanie Jegelka, Le Song, Csaba Szepesvari, Gang Niu, and Sivan Sabato (eds.), Proceedings of the 39th Inter- national Conference on Machine Learning, volume 162 of Proceedings of Machine Learning Research, pp. 17644–17655. PMLR, 17–23 Jul 2022. URL https://proceedings.mlr.press/v162/peng22b.html.
>
> [9] Arne Schneuing, Yuanqi Du, Charles Harris, Arian Jamasb, Ilia Igashov, Weitao Du, Tom Blun- dell, Pietro Li ́o, Carla Gomes, Max Welling, Michael Bronstein, and Bruno Correia. Structure- based drug design with equivariant diffusion models, 2023. URL https://arxiv.org/abs/2210.13695
>
> [10] Jiaqi Guan, Wesley Wei Qian, Xingang Peng, Yufeng Su, Jian Peng, and Jianzhu Ma. 3d equivariant diffusion for target-aware molecule generation and affinity prediction. In The Eleventh Interna- tional Conference on Learning Representations, 2023. URL https://openreview.net/forum?id=kJqXEPXMsE0

---

### Official Review · Reviewer_sxkU · 2023-10-29

**Soundness:** 3 good
**Presentation:** 3 good
**Contribution:** 3 good
**Rating:** 6
**Confidence:** 4

**Summary:**

In this work, the authors investigate the design space of equivariant diffusion generative model for de novo molecular design. The work applies EQGAT-diff, a modification from EQ-GAT as the score network. It mainly explores three aspects of equiavriant diffusion model: 1) fixed vs. time-dependent loss weighting 2) $\sigma$ vs. $x_0$ parametrization 3) discrete vs. continuous diffusion. Experiments on GEOM-QM9 and GEOM-DRUGS show a recipe for better de novo molecular design with equivariant diffusion model. By navigating through the design spaces, the model also achieves SOTA performance than previous baselines.

**Strengths:**

1. The work is well-motivated. It is valuable to investigate the design space of equivariant diffusion model and provide a general recipe.
2. The proposed EQGAT-diff achieves superior performance through navigating through the design space.
3. The paper is well-written and easy to follow.

**Weaknesses:**

1. The work proposes to navigate through the design space of equivariant diffusion model for de novo 3D molecule design. However, there are still other design choices that are omitted, e.g., different diffusion kernel and backbones. I know it's infeasible to exhaust the whole design space in one paper. I want to put a note here about the possible extensions.
2. It seems why EQ-GAT is chosen to model score network is missing.

**Questions:**

1. Will different settings (like discrete vs continuous) lead to significantly different training time till convergence as well as inference time?
2. In section 6, the authors use OMEGA to generate ground truth. However, the conformers from OMEGA may be inaccurate. Did the authors consider pretrain on DFT datasets like PCQM4Mv2 (>3M data)? Though it's much smaller than PubChem3D, the model may benefit from higher data quality.
3. Following the previous question, Fig. 2 shows that with sufficient training data, the advantage of pretrained model is trivial. In fact, in some metrics, the pretrained model is doing slightly worse than training from scratch. What could be the reason? Could it be related to OMEGA generated pretraining data?
4. One question about the big scope of this work is what the benefits of direct de novo 3D molecule generative model vs. two-stage model where first generate SMILES/graphs and then predict the conformers? I can see the advantages for structure-conditioned molecule design. However, this paper mostly investigate unconditional 3D molecule generation.
5. Torsional diffusion (https://arxiv.org/abs/2206.01729) introduces a GEOM-XL dataset including large molecules (>100 atoms). It may also be interesting to see how the proposed method transfer to such OOD datasets.

---

> ### Author Response · Authors · 2023-11-17
> **Answer to reviewer sxkU**
>
> ### Weaknesses
>
> >The work proposes to navigate through the design space of equivariant diffusion model for de novo 3D molecule design. However, there are still other design choices that are omitted, e.g., different diffusion kernel and backbones.
> I know it's infeasible to exhaust the whole design space in one paper. I want to put a note here about the possible extensions.
>
> ### Answer
> Indeed, the design space of (equivariant) diffusion models is vast. Still, we have covered many choices through the experimental ablations while using one backbone neural network architecture.
>
> Keeping the same network throughout the experiments is crucial for better comparison,  e.g., applying the $\epsilon$-parameterization vs. $x_0$-parameterization for Gaussian diffusion.
> Regarding the choice of diffusion kernels, for the discrete diffusion, there is an alternative way to include an additional mask/absorbing state token that can be regarded as an augmented feature for a node. Having such a token in the discrete diffusion kernel might enable practitioners to model the deletion of a node. For example, assuming that we sample $N$ nodes by design, all $N$ nodes need to be populated and filled with one atom class. Similar to mask text in language modeling with an absorbing state feature/token, the model could perform the reverse sampling trajectory and potentially assign nodes in the point cloud of the absorbing state feature. We did not explore this in our work. We will mention possible extensions in our updated manuscript.
>
> >It seems why EQ-GAT is chosen to model score network is missing.
>
> ### Answer
> We have chosen EQGAT architecture as the backbone model due to the requirement of employing an equivariant graph neural network and its superior performance on protein benchmark datasets while being fast and simple with little computational overhead.
> Additionally, we had the motivation to utilize an attention-based graph neural network, which is why we also performed initial experiments using the TorchMD-Net model by [1], which has shown good performance in predicting small molecule properties like the potential energy. However, we found using TorchMD-Net did not perform as well as EQGAT. In preliminary work, we also found that the "classic" diffusion backbone, EGNN [2] performed slightly worse. Hence, we used EQGAT for all our experiments and finally built EQGAT-diff.
>
>
> ### Questions
> >Will different settings (like discrete vs continuous) lead to significantly different training time till convergence as well as inference time?
>
> ### Answer
> We evaluated our models during training every 20 epochs. Here, we noticed that the model with $x_0$-parameterization EQGAT-disc-x0 and SNR(t) loss weighting converged significantly faster than its continuous counterparts with and without loss weighting. The inference times for both parameterizations are very similar to each other. However, we also noticed that when using EQGAT-disc-x0 with loss weighting, we can reduce the number of diffusion timesteps to 100 without sacrificing much molecule stability and validity, respectively. Surprisingly, we find that the performance of EQGAT-disc-x0 without loss weighting trained on 500 diffusion steps is worse than with loss weighting on 100. Thus, we see a five-fold decrease in inference time when using the proposed framework (Appendix Fig. 10).

---

> ### Author Response · Authors · 2023-11-17
> **Continued**
>
> >In section 6, the authors use OMEGA to generate ground truth. However, the conformers from OMEGA may be inaccurate. Did the authors consider pre-train on DFT datasets like PCQM4Mv2 (>3M data)? Though it's much smaller than PubChem3D, the model may benefit from higher data quality.
>
> ### Answer
> We utilized the PubChem3D dataset because of its large size and the availability of 3D conformers covering a relevant part of the chemical space so that we do not need to run any further calculations to generate the pre-training dataset. As the primary goal of pretraining was to learn general chemistry and how atoms are placed in 3D space, the level of theory and quality of the structures the model was trained on was not our main concern. In fact, we used the PCQM4Mv2 dataset and ran initial experiments on it but could not find any benefit in using it. In fact, we found that the level of theory and the presence of explicit hydrogens are not decisive for pre-training. More specifically, as shown in Fig. 2 in the main text and Fig. 8 in the Appendix, bond distance distributions do not match between GEOM-Drugs and PubChem3D, and
> although also explicit hydrogens are missing, fine-tuning is very effective even on small subsets of the target distribution with faster convergence.
>
> >Following the previous question, Fig. 2 shows that with sufficient training data, the advantage of pre-trained model is trivial. In fact, in some metrics, the pre-trained model is doing slightly worse than training from scratch. What could be the reason? Could it be related to OMEGA generated pretraining data?
>
> ### Answer
> As GEOM-Drugs provides a relatively large amount of training data (we used roughly 1.5M conformers for the entire dataset), it is reasonable that the advantage of the fine-tuned model would be reduced with increasing dataset size. Nevertheless, we did find that in terms of molecule stability and validity, the fine-tuned model consistently outperforms the model trained from scratch. Only the valency distribution is slightly worse, so we are not quite sure to which performance the reviewer is referring. As can be seen in the main text in Fig. 2 in the second row, this happens although the model needs to learn a significant distribution shift (in chemical space, but also in terms of the level of theory and software used to produce the structures; see answer above).
>
> >One question about the big scope of this work is what the benefits of direct de novo 3D molecule generative model vs. two-stage model where first generate SMILES/graphs and then predict the conformers? I can see the advantages for structure-conditioned molecule design. However, this paper mostly investigate unconditional 3D molecule generation.
>
> ### Answer
>
> Our aim has been to establish an accurate, robust, fast framework for direct 3D molecule generation before addressing advanced use cases. Nevertheless, to further underline the applicability and importance of our proposed model, we added results on QM property conditional and structure-based molecule design to the manuscript.
>
> >Torsional diffusion (https://arxiv.org/abs/2206.01729) introduces a GEOM-XL dataset including large molecules (>100 atoms). It may also be interesting to see how the proposed method transfer to such OOD datasets.
>
> ### Answer
>
> The Torsional Diffusion model is a 3D conformer generator that gradually changes torsion angles from a given input 3D structure (e.g., obtained through the RDKit). Hence, the molecule and its atom composition and bond topology do not change. In our scenario of de novo, unsupervised generative modeling, we do not see a direct way to evaluate the GEOM-XL dataset. However, as shown in Figure 3 of our manuscript, our model can sample larger molecules comprising 80-100 atoms but with some decreased performance regarding molecule stability. This is expected, as the diffusion model, which learns the distribution of atoms in a point cloud, was trained on a dataset with an average of around 45 atoms (see Figure 5 in the Appendix).
>
> We thank the reviewer for the feedback.
>
> A critical aspect of drug discovery is structure-based conditional molecule generation. Here the task is to sample molecules with specific quantum- and physicochemical properties in the presence of known protein receptors.
> To further highlight the importance of our findings, we provide results on structure-based de novo ligand generation. We evaluated EQGAT-diff on the Crossdocked dataset \cite{crossdocked} and found that SNR-t weighting and finetuning are crucial components confirming our previous results.
>
> We believe that the revision of the manuscript significantly improves the content and quality of our paper.
> Thus, we ask the reviewer to consider raising the score.

---

> > ### Author Response · Authors · 2023-11-17
> > **References**
> >
> > ### References
> >
> > [1] Philipp Thoelke and Gianni De Fabritiis. Equivariant transformers for neural network based molec-
> > ular potentials. International Conference on Learning Representations, 2022. URL https://openreview.net/forum?id=zNHzqZ9wrRB
> >
> > [2] E (n) equivariant graph neural networks
> > VG Satorras, E Hoogeboom, M Welling
> > International conference on machine learning, 9323-9332

---

> ### Comment · Reviewer_sxkU · 2023-11-20
> **Thanks for the reply**
>
> I thank the authors for answering my questions and adding experiments of conditional generation. I stay positive about the work.

---

### Official Review · Reviewer_AUdU · 2023-11-01

**Soundness:** 3 good
**Presentation:** 3 good
**Contribution:** 3 good
**Rating:** 8
**Confidence:** 4

**Summary:**

This paper describes EQGAT-diff, an improvement to the EQGAT architecture (a 3D small-molecule generator) that includes diffusion. The paper explores a couple of design choices in the formulation of the diffusion part of this updated method, such as the addition or not of the truncated SNR loss term, and the inclusion of a discrete loss term for the molecular graph construction. The paper shows that their improved model is better than MiDi, the method that introduced diffusion on the molecular graphs jointly with the coordinate reconstruction.

**Strengths:**

The method is a significant advance compared to the EQGAT method. The exploration of the design choices is reasonable and of some interest, even though the results are not surprising based on the existing literature or simple logic.

The introduction and method is well written and clear. Aspects of sections 5 and 6 could improve, in particular the figures.

The datasets that the authors use are standard in this line of study, and they have well know limitations, and it is surprising to me that the community has not yet moved to better data, which should be reasonably easy as QM calculations are not that hard to perform.

The overall significance of this work appears more specialized than a usual contribution to ICLR, however, the paper would be of some interest to the growing number of drug discovery researchers that apply ML to small molecules.

**Weaknesses:**

The novelty of the approach is more specialized than a typical ICLR as it is more of a collection of good ideas from a narrow field of study. The authors didn't discuss or try to draw ideas from recent efforts in generative diffusion models for images or for protein structure prediction that may have strong analogies and lessons to the study of small molecules as well.

The fonts in all the figure labels are unreadably (unpublishably?) small. If space is the main concern, then there certainly exist small chunks of text (e.g. the last paragraph of section 4) and some figure panels (e.g. most from Fig 2) that could move to the supplement, and some parts that could be rewritten concisely (e.g. intro to sec 5, which could also be clarified as it was not clear what "these three aspects" meant in paragraph 2, top of page 5).


The dataset is a weakness for this paper and most other papers in this area.  I disagree with the premise of the authors that the availability of molecular data is not as abundant, though I agree that the community has been stuck on trivial and no longer meaningful datasets (it feels similar to ML in vision prior to imagenet). The authors mention the catalogue of Enamine Real in the supplement which scales to the 10s of billions of molecular graphs (albeit limited in complexity, but there are also computational enumerations of valid molecules that can be much bigger), the published patent literature probably scales to the multiple 10s of millions.  QM methods of relatively reasonable accuracy exist that are not exceedingly expensive and datasets of 10s of millions of published 3D structures with simpler methods also exist.

**Questions:**

The paper uses E3 symmetries which includes the reflection group and can therefore mix chiralities, in principle changing the input distribution of molecular graphs.  The authors seem to not address the possible distribution shifts due to chiral transformations.

Small molecules with 3D coordinates often have specific chirality.  Although this chirality is specified in the full set of 3D positions, the authors model is using E3 symmetries in its core so I wonder if it will mix chiral molecule in ways that would result in a distribution shift.  Have the authors checked the chirality of the generated molecular distributions by keeping say only one kind of stereo isomer and validating that their model would construct both? Would implementing the SE3 group allow the authors to also including chiral flags on the bonds, in the way that shown in the typical 2D representations of molecular graphs?

It was not clear to me why the inclusion of the eps-parameterization does not use a composite loss as did the x0-parameterization.  Is it not a priori clear a priori that the addition of a categorical loss for categorical data will outperform a mean squared error on categorical data?  I may be misunderstanding something simple here, so perhaps you can explain here and possibly reformulate the early parts of section 5.

The lack of the reproduction of the MiDi code benchmarks that the authors report in the supplement is interesting, though it probably would not change the results in the paper in a qualitative way.  Did the authors contact the authors of MiDi to resolve this issue and did they receive any feedback on that question?

The observation by the authors regarding deteriorating performance upon pretraining on explicit hydrogen on a larger molecular set is an interesting point that gets buried in the supplement. Is there a generalizable lesson and perhaps a better/different way to overcome this problem than dropping the hydrogens, or is this observation too close to the level of noise?

---

> ### Author Response · Authors · 2023-11-17
> **Answer to reviewer AUdU**
>
> ### Weaknesses
>
> > The novelty of the approach is more specialized than a typical ICLR as it is more of a collection of good ideas from a narrow field of study. The authors didn't discuss or try to draw ideas from recent efforts in generative diffusion models for images or for protein structure prediction that may have strong analogies and lessons to the study of small molecules as well.
>
> ### Answer
>
> We are unsure which specific analogies and lessons the reviewer is referring to.
> We also explored DDIM sampling, known for enhancing inference/sampling time in diffusion models trained via the standard DDPM procedure in image processing. The difference between DDIM and DDPM lies in the sampling algorithm, which we believe could also be applied in our setting of molecular data. However, our best-performing scenario utilizes the $ x_0$-parameterization to preserve the correct data modalities for coordinates, atoms, and bond features. Thus, applying DDIM directly to discrete-valued data modalities is not straightforward. We restricted DDIM to continuous coordinate updates, while discrete-valued data modalities follow the approach outlined by [1] and explained in our Appendix (A.1.2). Tab. 11[edited] in the Appendix compares the evaluation performance of our base models when generating samples using DDIM or DDPM for varying numbers of reverse sampling steps (500, 250, 167).
> Given that all models underwent training with $T=500$ discretized timesteps, we conducted DDIM sampling every 2 or 3 steps of the reversed trajectory starting from index 500. Notably, we observed that employing DDIM did not enhance the quality of molecule generation with fewer sampling steps (250 or 166) compared to the 500 steps the models were trained on. Consequently, we opted not to include the DDIM algorithm in our submitted manuscript.
> We performed additional experiments and trained EQGAT$_{disc}^{x_0}$ with $T=100$ time steps using the uniform and truncated SNR(t) loss weighting. The rationale behind these experiments is to assess whether or not the number of steps could be reduced using uniform or our proposed loss weighting scheme in terms of performance, enabling faster inference time. Indeed, we observe that the model trained with loss weighting over $T=100$ steps performs better than the model trained with $T=500$ steps using uniform loss weighting, as illustrated in Fig. 10 in the Appendix of the updated manuscript. Although we found the model trained with $T=500$ steps still performs better than the one trained with $T=100$, the difference is relatively marginal compared to the five-fold gain in inference speed. Thus, our results again show that using the proposed model with loss weighting is crucial.
>
>
> >The fonts in all the figure labels are unreadably (unpublishably?) small. If space is the main concern, then there certainly exist small chunks of text (e.g. the last paragraph of section 4) and some figure panels (e.g. most from Fig 2) that could move to the supplement, and some parts that could be rewritten concisely (e.g. intro to sec 5, which could also be clarified as it was not clear what "these three aspects" meant in paragraph 2, top of page 5).
>
> ### Answer
> We updated all our figures with increased font size and line widths for improved readability and moved details from the end of Section 4 into the Appendix as suggested by the reviewer. Furthermore, we clarified the introduction of Section 5 as marked in blue color.

---

> ### Author Response · Authors · 2023-11-17
> **Continued**
>
> >The dataset is a weakness for this paper and most other papers in this area. I disagree with the premise of the authors that the availability of molecular data is not as abundant, though I agree that the community has been stuck on trivial and no longer meaningful datasets (it feels similar to ML in vision prior to imagenet). The authors mention the catalog of Enamine Real in the supplement which scales to the 10s of billions of molecular graphs (albeit limited in complexity, but there are also computational enumerations of valid molecules that can be much bigger), the published patent literature probably scales to the multiple 10s of millions. QM methods of relatively reasonable accuracy exist that are not exceedingly expensive and datasets of 10s of millions of published 3D structures with simpler methods also exist.
>
> ### Answer
>
> We disagree on the weakness of GEOM-Drugs, as this dataset covers an interesting area of the chemical space using a well-established semi-empirical level of theory (xTB-GFN2) and has previously been hard to learn (EDM and similar models failed to learn the data distribution).
> In principle, there is a large amount of publicly available SMILES strings and molecular graphs. But, for structure-based tasks or tasks that involve experimental measurements, the data is by far not as abundant. This is precisely why we show that pre-training on large databases for 3D conformers like PubChem3D can help with that significantly, even though they have been generated only with empirical force fields.
> We show that for pre-training, the level of theory and the presence of explicit hydrogens is not decisive. Specifically, as shown in Fig. 2 in the main text and Fig. 8 in the Appendix, bond distance distributions do not match between GEOM-Drugs and PubChem3D. Although explicit hydrogens are also missing, fine-tuning is very effective even on small subsets of the target distribution with faster convergence.
>
> ### Questions
> >The paper uses E3 symmetries, which include the reflection group and can therefore mix chiralities, in principle changing the input distribution of molecular graphs. The authors seem to not address the possible distribution shifts due to chiral transformations.
> Small molecules with 3D coordinates often have specific chirality. Although this chirality is specified in the full set of 3D positions, the authors model is using E3 symmetries in its core so I wonder if it will mix chiral molecule in ways that would result in a distribution shift. Have the authors checked the chirality of the generated molecular distributions by keeping say only one kind of stereo isomer and validating that their model would construct both? Would implementing the SE3 group allow the authors to also including chiral flags on the bonds, in the way that shown in the typical 2D representations of molecular graphs?
>
> ### Answer
> Our model does not consider chirality for single molecules since the unconditional distribution is symmetric under reflection. The reflection symmetry is only broken for use cases where an external reference system, e.g., a target protein, is introduced. When this is encoded in the model condition, for example, when the model is to be trained to generate a 3d ligand into a 3d pocket, the chirality of the molecule is correctly reflected in the conditional distribution.
> If one aims to represent chirality directly, we believe implementing a chirality-sensitive function on torsion angles would be possible, as described in [2]. For diffusion-based de-novo design, however, it is not straightforward to enumerate the corresponding torsion angles since the molecular graph changes over time.

---

> ### Author Response · Authors · 2023-11-17
> **Continued**
>
> >It was not clear to me why the inclusion of the eps-parameterization does not use a composite loss as did the x0-parameterization. Is it not a priori clear a priori that the addition of a categorical loss for categorical data will outperform a mean squared error on categorical data? I may be misunderstanding something simple here, so perhaps you can explain here and possibly reformulate the early parts of section 5.
>
> ### Answer
> The diffusion model aims in minimizing the KL-divergence $D_\text{KL}(q(x_{t-1}|x_t, x_0) || p_\theta(x_{t-1}|x_t) )$, while for continuous data $x$, the tractable reverse distribution $q(x_{t-1}|x_t, x_0)$ is a multivariate normal distribution [Ho et al. 2020], and for discrete data $x$, a categorical distribution [Hoogeboom et al. 2021, Austin et al. 2021].
> The best choice to implement $p_\theta$ is to predict the corresponding distribution parameters of $q(x_{t-1}|x_t, x_0)$, which, for the continuous case, leads to predicting the mean of $q(x_{t-1}|x_t, x_0)$ or equivalently the noise $\epsilon$ that was used when perturbing $x_0$ to $x_t$, overall minimizing a mean squared error loss, since the Gaussian includes the quadratic term.
> In the discrete case, the \textit{derivation} for the objective function of the KL-divergence results in the cross-entropy loss $L_\text{CE} = -\sum_{k}^K p_k \log(\hat{p}_k)$ as discussed in [1].
> In both cases, the inherent connection to maximum likelihood and assumption of the data type results in the mean-squared error or cross-entropy loss.
> However, it is true that one could utilize a mean squared error on the predicted class probabilities resulting in the loss term $\sum_k (p_k - \hat{p}_k)^2$. Since the softmax cross-entropy loss has been established in the supervised learning formulation, especially in natural language and image processing, we believe it is the first choice to implement, next to our initial motivation to use the proper loss function when deriving from the KL divergence minimization.
>
> >The lack of the reproduction of the MiDi code benchmarks that the authors report in the supplement is interesting, though it probably would not change the results in the paper in a qualitative way. Did the authors contact the authors of MiDi to resolve this issue and did they receive any feedback on that question?
>
> ### Answer
> We contacted the authors of MiDi, but we did not get enlightening answers to our questions. As we used not only the provided checkpoint but also re-trained the MiDi model using the provided code and could in both cases not verify or reproduce the results, we assume that the authors made a mistake in reporting. Nevertheless, it is true there are no qualitative changes for the MiDi publication.
>
> >The observation by the authors regarding deteriorating performance upon pretraining on explicit hydrogen on a larger molecular set is an interesting point that gets buried in the supplement. Is there a generalizable lesson and perhaps a better/different way to overcome this problem than dropping the hydrogens, or is this observation too close to the level of noise?
>
> ### Answer
> We do not have a clear answer for the inferior performance when pretraining on PubChem3D with explicit hydrogens. One hypothesis is that the diffusion model has a simpler task during finetuning to learn to add hydrogens onto a molecule as opposed to relearn a distribution shift of the attached hydrogens.
> To investigate this, we subsampled 1M molecules from PubChem3D and GEOM-Drugs and enumerated all bonds that involve hydrogens. We computed distances and noticed that the hydrogen-oxygen distances distribution in PubChem3D seems to have a smaller variance compared to GEOM-Drugs as illustrated in Fig. 8 in the Appendix of the updated manuscript.
> Nevertheless, our results suggest that the diffusion model can effectively learn a distribution shift underpinning the importance of pre-training on large 3D databases independent of the level of theory and the presence of explicit hydrogens. The pre-trained model yields superior performance while accurately learning the underlying data distribution faster from less data of the target distribution (Fig. 2, Appendix Fig. 8).

---

> ### Author Response · Authors · 2023-11-17
> **Continued**
>
> We conducted new experiments, including the assessment of improved sampling mechanisms. We trained our diffusion models with fewer timesteps and observed that our proposed loss weighting greatly helps in maintaining sample quality while reducing the number of diffusion steps and hence inference time.
> Furthermore, we added new experiments that go in the direction of conditional molecule design, which is of more relevance to the pharmaceutical research area. We extended the framework to also work with classifier-guidance (Appendix Tab. 8) as well as on structure-based ligand generation (Tab. 4). Importantly, we see that our framework, including fine-tuning, helps in learning on smaller datasets for protein-ligand interaction, confirming and reinforcing the importance of our findings.
>
> Thus, we ask the reviewer to consider raising their score.
>
> [1] Jacob Austin, Daniel D. Johnson, Jonathan Ho, Daniel Tarlow, and Rianne van den Berg. Structured
> denoising diffusion models in discrete state-spaces. In A. Beygelzimer, Y. Dauphin, P. Liang, and
> J. Wortman Vaughan (eds.), Advances in Neural Information Processing Systems, 2021. URL
> https://openreview.net/forum?id=h7-XixPCAL
>
> [2] Keir Adams, Lagnajit Pattanaik, and Connor W. Coley. Learning 3d representations of molecular
> chirality with invariance to bond rotations. In International Conference on Learning Representa-
> tions, 2022. URL https://openreview.net/forum?id=hm2tNDdgaFK
>
> [3] Emiel Hoogeboom, V ́ıctor Garcia Satorras, Cl ́ement Vignac, and Max Welling. Equivariant diffu-
> sion for molecule generation in 3D. In Kamalika Chaudhuri, Stefanie Jegelka, Le Song, Csaba
> Szepesvari, Gang Niu, and Sivan Sabato (eds.), Proceedings of the 39th International Confer-
> ence on Machine Learning, volume 162 of Proceedings of Machine Learning Research, pp.
> 8867–8887. PMLR, 17–23 Jul 2022. URL https://proceedings.mlr.press/v162/hoogeboom22a.html
>
> [4] Jonathan Ho, Ajay Jain, and Pieter Abbeel. Denoising diffusion probabilistic models. In
> H. Larochelle, M. Ranzato, R. Hadsell, M.F. Balcan, and H. Lin (eds.), Advances in Neu-
> ral Information Processing Systems, volume 33, pp. 6840–6851. Curran Associates, Inc.,
> 2020. URL https://proceedings.neurips.cc/paper_files/paper/2020/file/4c5bcfec8584af0d967f1ab10179ca4b-Paper.pdf

---

> > ### Author Response · Authors · 2023-11-22
> > **Second comment by the authors**
> >
> > We thank the reviewer for the initial feedback.
> > We have improved our work in response to the comments, with added experiments on conditional molecule generation, e.g., ligand binding to a receptor in Sec. 8 of our updated manuscript. Here, we show that our proposed framework, initially introduced in the context of unconditional generative modelling, can seamlessly transfer to structure-based tasks again showing state-of-the-art performance.
> >
> > Following the reviewer's comment, we also believe that our work will be of significant interest for a growing proportion of ICLR participants, as machine learning in drug discovery is gaining more and more interest.
> > But, we are happy to clarify any remaining concerns the reviewer may have.
> > Our additional experiments on conditional generation consolidate our initial profound findings and thus we would ask the reviewer to consider raising the score.

---

### Official Review · Reviewer_XyXs · 2023-11-01

**Soundness:** 2 fair
**Presentation:** 2 fair
**Contribution:** 2 fair
**Rating:** 6
**Confidence:** 3

**Summary:**

This paper explores the design space of equivariant diffusion-based generative models for de novo 3D molecule generation, including various parameterizations, loss weightings and data modalities. The authors then introduce EQGAT-Diff, which can achieve SOTA results in shorter training time and with less trainable parameters.

**Strengths:**

This is an empirical paper. The experiments part is solid: various metics are computed, many ablation studies are performed and standard deviations are reported.  The explored design is helpful for future model design in the community of ML + chemistry.

**Weaknesses:**

- The novelty is limited. Most explored designs are easy to think about, like time-dependent loss weight, modeling discrete or continuous atom/bond types, parameterizing to match noise or x0, etc. The authors didn't come up with new designs.
- It would be better to summarize useful designs / helpful findings clearly somewhere.
- One important baseline MolDiff [1] is not compared with, although it has been cited.

[1] Peng, X., Guan, J., Liu, Q., & Ma, J. MolDiff: Addressing the Atom-Bond Inconsistency Problem in 3D Molecule Diffusion Generation. ICML 2023.

**Questions:**

N/A

---

> ### Author Response · Authors · 2023-11-17
> **Answer to reviewer XyXs**
>
> ### Weaknesses
>
> > The novelty is limited. Most explored designs are easy to think about, like time-dependent loss weight, modeling discrete or continuous atom/bond types, parameterizing to match noise or x0, etc.
>
> ### Answer
>
> Our extensive analysis and benchmark with many ablations of different design choices on our EQGAT-diff model provides a profound contribution to the machine learning community, especially for researchers with applications in the bio-chemistry domain. Some implemented design choices have been initially proposed in the image and language processing community. Nevertheless, We argue that our work still contains highly relevant in-depth experimentation. We set a new state-of-the-art emphasizing important design choices crucial for successfully training and validating diffusion models on molecular data. Our findings lead to faster training convergence and increased inference speed while providing significantly higher stability and validity on larger molecular structures.
>
> > The authors didn't come up with new designs.
> It would be better to summarize useful designs / helpful findings clearly somewhere.
>
> ### Answer
> As mentioned in the previous answer, while our work does not provide a completely new method for 3D molecule design, we concentrate on proper experimentation of well-known theoretical findings. In the updated manuscript, we outlined the summary of design choices in the introduction.
>
> > One important baseline MolDiff [1] is not compared with, although it has been cited.
>
> ### Answer
>
> MolDiff utilizes a different pre-processing and splitting of the GEOM-Drugs dataset compared to EDM and MiDi while using an enhanced post-processing pipeline that fixes valency and aromaticity for generated molecules before computing validity and success rate. We adopted the data and evaluation pipeline from EDM and MiDi, which is not directly comparable to the results presented by MolDiff.
>
> In the updated manuscript, we added results following the evaluation pipeline from MolDiff's GitHub repository (commit \#861e82c) and evaluated our generated samples. We report the mean validity and mean success rate in Tab. 10 [edited] in the Appendix.
>
> Our proposed EQGAT-diff model achieves superior performance over MolDiff in generating chemically valid molecules, although MolDiff additionally utilizes uncertainty guidance via a separate bond-classifier model.
>
> We have carefully considered and addressed the concerns you raised. Building on well-known technical facts, we developed a state-of-the-art framework based on thorough experimental results. We considered many novel aspects in diffusion-based 3D modeling, like additional chemical features, denoising diffusion pre-training, and timestep-depending loss weighting. Our model improved upon standard metrics in this domain by large margins compared to established diffusion models like EDM, MiDi, and MolDiff. Furthermore, we have extended our framework to include not only classifier-guided molecular design but also structure-based ligand generation and find that our findings are also applicable in these situations, underpinning the importance of this work
>
> Thus, we ask the reviewer to consider raising the score.

---

> ### Author Response · Authors · 2023-11-22
> **Second comment by the authors**
>
> We thank the reviewer for the initial feedback.
> We have improved our work in response to the raised concerns, clarifying our contributions, highlighting the best setting for model design in 3D molecule diffusion and included new empirical evidence that our proposed framework is broadly applicable also in the conditional generation setting.
> We are happy to clarify any remaining concerns the reviewer may have.
>
> Given the additional conditional generation experiments that support our initial profound findings, we would ask the reviewer to consider increasing the score
>
> ***

---

### Author Response · Authors · 2023-11-17
**General answer**

We thank all reviewers for taking their time and providing insightful comments. We address the specific concerns in detailed responses to each review below. Here, we summarize the main issues and highlight the significant differences between the original and revised submissions.

### Task Relevance
Our study extensively examines various design choices for diffusion-based 3D molecular data generation. Initially, we focused on unconditional 3D molecule generation to gain relevant insights into essential design decisions, such as tailoring fast and powerful equivariant deep learning models, incorporating categorical diffusion on discrete data modalities, and a task-specific loss weighting function. Our findings show new state-of-the-art results by a large margin while being significantly faster in training and inference (Appendix Fig. 9 and Fig. 10). Thus, this work helps build the foundation for further investigations in generative chemistry.

A critical aspect of it is, e.g., structure-based conditional molecule generation. Here, sampling molecules with specific quantum- and physicochemical properties in the presence of known protein receptors necessitates accurate, reliable, and fast 3D molecular modeling.

To further highlight the importance of our findings, we provide results on structure-based de novo ligand generation. Here, we evaluated EQGAT-diff on the Crossdocked dataset [3] and found that SNR-t weighting and finetuning are crucial components confirming our previous results (Tab. 4)

For conditional molecule design, we can use a trained unconditional EQGAT-diff model and classifier-guidance, as proposed in [1], to steer the generation of samples using the gradient of an external classifier/regressor during the reverse sampling trajectory from noise to data. As a proof of concept, we explored classifier-guidance to generate molecules optimizing for low/high polarizability, showing promising results (Appendix A.8, Tab.9) [edited]

### Technical Novelty
In this work, we provide a profound contribution by rigorous experimental design containing many important ablations.
We think that in science, experimental validation and exploration are at least as important as technical novelty. Especially in ML, we find that proper evaluation by in-depth experimentation and ablation is often harmed when focusing on delivering theoretical or architectural novelty. This work's main idea and driver was the belief that current state-of-the-art 3D diffusion models were not sufficiently examined, resulting in significant flaws, e.g., in the design of larger molecular structures. Thus, we focused on a concise experimental investigation of well-known theoretical findings. We show that by exploring the diffusion design space, it is possible to vastly increase the model's generative performance - e.g., compared to the latest SOTA (MiDi) [2], our model increases the validity of sampled molecules on average by roughly 25\%; we increase the molecular stability (3D metric) by roughly 240\% compared to EDM [4]; we increase the inference speed by a factor of 5 compared to MiDi and EDM.

### Code Availability
The code will be made public as soon as our internal clearance is finished in the upcoming days.
We have developed a code base that is easy to use and allows full reproducibility of the underlying work.

# References
[1] Prafulla Dhariwal and Alexander Quinn Nichol. Diffusion models beat GANs on image synthesis.
In A. Beygelzimer, Y. Dauphin, P. Liang, and J. Wortman Vaughan (eds.), Advances in Neu-
ral Information Processing Systems, 2021. URL https://openreview.net/forum?id=AAWuCvzaVt.

[2] Clement Vignac, Nagham Osman, Laura Toni, and Pascal Frossard. Midi: Mixed graph and 3d
denoising diffusion for molecule generation. In Danai Koutra, Claudia Plant, Manuel Gomez
Rodriguez, Elena Baralis, and Francesco Bonchi (eds.), Machine Learning and Knowledge
Discovery in Databases: Research Track - European Conference, ECML PKDD 2023, Turin,
Italy, September 18-22, 2023, Proceedings, Part II, volume 14170 of Lecture Notes in Com-
puter Science, pp. 560–576. Springer, 2023. doi: 10.1007/978-3-031-43415-0\ 33. URL
https://doi.org/10.1007/978-3-031-43415-0_33

[3] Three-Dimensional Convolutional Neural Networks and a Cross-Docked Data Set for Structure-Based Drug Design
Paul G. Francoeur, Tomohide Masuda, Jocelyn Sunseri, Andrew Jia, Richard B. Iovanisci, Ian Snyder, and David R. Koes
Journal of Chemical Information and Modeling 2020 60 (9), 4200-4215DOI: 10.1021/acs.jcim.0c00411

[4] Emiel Hoogeboom, V ́ıctor Garcia Satorras, Cl ́ement Vignac, and Max Welling. Equivariant diffu-
sion for molecule generation in 3D. In Kamalika Chaudhuri, Stefanie Jegelka, Le Song, Csaba
Szepesvari, Gang Niu, and Sivan Sabato (eds.), Proceedings of the 39th International Confer-
ence on Machine Learning, volume 162 of Proceedings of Machine Learning Research, pp.
8867–8887. PMLR, 17–23 Jul 2022. URL https://proceedings.mlr.press/v162/hoogeboom22a.html

---

> ### Author Response · Authors · 2023-11-21
> **General answer (2) - New results and updated manuscript**
>
> Dear reviewers,
>
> we included additional results evaluating our model on structure-based de novo ligand design and compared against two related works, namely DiffSBDD [1] and TargetDiff [2] which both are also diffusion-based generative models providing state-of-the-art performance in this domain.
>
> We used our best-performing model, pre-trained on PubChem3D and fine-tuned on the CrossDocked dataset, to generate ligands for docking evaluation. Following the authors of [1], [2] and [3], we sampled 100 ligands per target from the test dataset and calculated the docking score using QVina.
>
> As shown in Tab. 5 in the updated manuscript (and also here below), our model outperforms both TargetDiff [2] and DiffSBDD [3] in terms of mean docking score over all generated ligands, while showing a higher synthesizability and diversity in the samples.
>
> ***
>
> | Model                           | Vina (All) $\downarrow$       | Vina (Top-10\%) $\downarrow$  | QED $\uparrow$               | SA $\uparrow$                | Lipinski $\uparrow$ | Diversity $\uparrow$         |
> |---------------------------------|-------------------------------|-------------------------------|------------------------------|------------------------------|---------------------|------------------------------|
> | EQGAT-diff| $\mathbf{-7.423}_{\pm{2.33}}$ | -9.571$_{\pm{2.14}}$          | $\mathbf{0.522}_{\pm{0.18}}$ | $\mathbf{0.697}_{\pm{0.20}}$ | 4.66$_{\pm{0.72}}$  | $\mathbf{0.742}_{\pm{0.07}}$ |
> | TargetDiff                      | -7.318$_{\pm{2.47}}$          | $\mathbf{-9.669}_{\pm{2.55}}$ | 0.483$_{\pm{0.20}}$          | 0.584$_{\pm{0.13}}$          | 4.594$_{\pm{0.83}}$ | 0.718$_{\pm{0.09}}$          |
> | DiffSBDD-cond                   | -6.950$_{\pm{2.06}}$          | -9.120$_{\pm{2.16}}$          | 0.469$_{\pm{0.21}}$          | 0.578$_{\pm{0.13}}$          | 4.562$_{\pm{0.89}}$ | 0.728$_{\pm{0.07}}$          |
>
> ***
>
> With that,  we have considerably reinforced the relevance of our findings and the importance of our proposed framework.
> Thus, we would ask the reviewers to consider raising their scores.
>
> ### References
> [1]  Xingang Peng, Shitong Luo, Jiaqi Guan, Qi Xie, Jian Peng, and Jianzhu Ma. Pocket2Mol: Efficient molecular sampling based on 3D protein pockets. In Kamalika Chaudhuri, Stefanie Jegelka,
> Le Song, Csaba Szepesvari, Gang Niu, and Sivan Sabato (eds.), Proceedings of the 39th Inter-
> national Conference on Machine Learning, volume 162 of Proceedings of Machine Learning Research, pp. 17644–17655. PMLR, 17–23 Jul 2022. URL https://proceedings.mlr.press/v162/peng22b.html.
>
> [2] Arne Schneuing, Yuanqi Du, Charles Harris, Arian Jamasb, Ilia Igashov, Weitao Du, Tom Blun-
> dell, Pietro Li ́o, Carla Gomes, Max Welling, Michael Bronstein, and Bruno Correia. Structure-
> based drug design with equivariant diffusion models, 2023. URL https://arxiv.org/abs/2210.13695
>
>
> [3] Jiaqi Guan, Wesley Wei Qian, Xingang Peng, Yufeng Su, Jian Peng, and Jianzhu Ma. 3d equivariant
> diffusion for target-aware molecule generation and affinity prediction. In The Eleventh Interna-
> tional Conference on Learning Representations, 2023. URL https://openreview.net/forum?id=kJqXEPXMsE0

---

> > ### Author Response · Authors · 2023-11-23
> > **Link to anonymous repository**
> >
> > We are currently waiting for approval to open-source our code to the public.
> > Since we are in favour of open science and sharing code, we have created an empty repository as placeholder and anonymized it at https://anonymous.4open.science/r/eqgat-diff-DC5D/ .
> > We will include and update our code base with necessary files to re-run experiments and sampling to share with the reviewers as soon as possible. Since the rebuttal period ends very soon, we wanted to use the last opportunity to edit and share the link of the anonymous repository with you.

---

### Comment · Area_Chair_sGPg · 2023-11-21

Dear Reviewers:

Given the varying perspectives on the significance of this paper, I am grateful for your willingness to engage with the authors for any necessary future clarifications.

Thank you,

AC

---

> ### Comment · Area_Chair_sGPg · 2023-12-05
> **Discussion is needed given the recent changes in scores**
>
> Dear Reviewers,
>
> There has been a notable recent shift in reviewers' sentiment that now makes the paper worth discussing. To help calibrate the diverging scores, additional feedback from reviewers highlighting the specific aspects that influenced their change in sentiment would be valuable. If available, I would appreciate any further details that could aid in a more comprehensive evaluation.
>
> Thanks,
>
> AC

---

> > ### Comment · Reviewer_AUdU · 2023-12-05
> > **Reasons for increasing score**
> >
> > The authors addressed my concerns fully by demonstrating conditional generation of small molecules (both the concerns on SE3 symmetries and on relevance) they commented on additional advances that didn't seem to work in their case, they tried harder to explain the deterioration with the inclusion of hydrogens, and they fixed all the small annoyances in the figures.  I agree with the authors that the paper now seems worthy of ICLR.  I remain skeptical of GEOM-Drugs as a pretraining solution, but in this case the paper is worthwhile despite using this particular training dataset.

---

### Meta-Review · Area_Chair_sGPg · 2023-12-08

**Metareview:**

This paper introduces EQGAT-diff, a framework for de novo 3D molecular generation that performs well in terms of speed and accuracy compared to other models. The authors highlight its potential contributions to drug discovery through 3D-based generative modeling, especially in conditional molecular modeling. While the initial scores were low, reviewers' sentiments significantly improved after reading the authors' rebuttals. Notably, concerns raised by a reviewer regarding SE3 symmetries and relevance have been addressed with additional results on the conditional generation of small molecules.

**Justification For Why Not Higher Score:**

The reviewers highlight the empirical nature of the paper and emphasize its limited novelty, noting that the explored designs are perceived as straightforward and lacking innovation. The absence of new and distinctive designs is pointed out, and there is a suggestion to clearly summarize useful findings.

Furthermore, one reviewer raises concerns about the inherent meaningfulness of 3D molecular generation for drug discovery. The suggestion is to emphasize potential limitations and stress the necessity for practical utility to enhance the paper's relevance.

**Justification For Why Not Lower Score:**

The paper received a positive evaluation due to its strong experimental foundation, featuring robust experiments, comprehensive metrics, numerous ablation studies, and reported standard deviations. The abundance of design exploration is highlighted as a key factor in not assigning a lower score.

---

### Decision · Program_Chairs · 2024-01-16

Accept (poster)